



# A spectral library for laser-induced fluorescence analysis as a tool for rare earth element identification

Margret C. Fuchs[1], Jan Beyer[2], Sandra Lorenz[1], Suchinder Sharma[1, 2], Axel D. Renno[1], Johannes Heitmann[2], and Richard Gloaguen[1]

[1]Helmholtz-Zentrum Dresden-Rossendorf, Helmholtz Institute Freiberg for Resource Technology, Chemnitzer Str. 40, 09599 Freiberg, Germany
[2]Institute of Applied Physics, TU Bergakademie Freiberg, Leipziger Strasse 23, 09599 Freiberg, Germany

**Correspondence:** Margret C. Fuchs (m.fuchs@hzdr.de)

**Abstract.**

With the recurring interest on rare-earth elements (REE), laser-induced fluorescence (LiF) may provide a powerful tool for their rapid and accurate identification at different stages along their value chain. Applications to natural materials such as rocks could complement the spectroscopy-based toolkit for innovative, non-invasive exploration technologies.
However, the diagnostic assignment of detected emission lines to individual REE remains challenging, because of the complex composition of natural rocks in which they can be found. The resulting mixed spectra and the large amount of data generated demand for automated approaches of data evaluation, especially in mapping applications such as drill core scanning. LiF reference data provide the solution for robust REE identification, yet they usually remain in the form of tables of published emission lines. We show that a complete reference spectra library could open manifold options for
innovative automated analysis.

We present a library of high-resolution LiF reference spectra using the Smithsonian rare-earth phosphate standards for electron microprobe analysis. We employ three standard laser wavelengths (325 nm, 442 nm, 532 nm) to record representative spectra in the UV-visible to near-infrared spectral range (340 - 1080 nm). Excitation at all three laser wavelengths yielded characteristic spectra with distinct REE-related emission lines for $EuPO_4$, $TbPO_4$, $DyPO_4$ and $YbPO_4$. In the
other samples, the high-energy excitation at 325 nm caused unspecific, broadband defect emissions. Here, lower energy laser excitation showed successful for suppressing non-REE-related emission. At 442 nm excitation, REE-reference spectra depict the diagnostic emission lines of $PrPO_4$, $SmPO_4$ and $ErPO_4$. For $NdPO_4$ and $HoPO_4$ most efficient excitation was achieved with 532 nm. Our results emphasise on the possibility of selective REE excitation by changing the excitation wavelength according to the suitable conditions for individual REEs. Our reference spectra provide a database
for transparent and reproducible evaluation of REE-bearing rocks. The LiF spectral library is available at zenodo.org and the registered DOI: 10.5281/zenodo.4054606 (Fuchs et al., 2020). It gives access to traceable data for manifold further studies on comparison of emission line positions, emission line intensity ratios and splitting into emission line sub-levels or can be used as reference or training data for automated approaches of component assignment.



## 1 Introduction

The exploding demand for rare-earth elements (REE) for the high-tech industry, e-mobility and the energy transition justifies the need for efficient detection methods all along the value chain, especially in raw material exploration and recycling, but also in science and processing or production monitoring (e.g., National Research Council, 2008; Lima and Filho, 2015; Barakos et al., 2016; European Commission, 2014, 2018). Spectroscopy-based methods are of paramount importance in overcoming time- and cost-intensive exploration routines in a world of depleting, increasingly complex and

more remote raw material deposits.

Reflectance spectroscopy has shown an immense potential for fast, non-invasive mineral mapping. Several studies particularily demonstrated advances in reflectance hyperspectral imaging (in the following abbreviated as HSI) of REEs in natural materials from close-range scanning of drill-cores and outcrops or from drone-borne data of larger-scale areas (e.g., Boesche et al., 2015; Turner, 2015; Neave et al., 2016; Zimmermann et al., 2016; Booysen et al., 2019). However,

an absorption feature-based identification relies in most application cases on single REEs (e.g. Nd, Pr, Sm), which then serve as pathfinders for the other REEs based on strong geochemical similarities and consequently, assuming generic relationships during geological processes. Key to result validation are reference spectra. Established spectral libraries such as from the USGS (Kokaly et al., 2017) give access to the necessary reference data for automated identification routines. This trend is also recognised by Fasnacht et al. (2019), who released a new library explicitly dedicated to serve

for advanced automated data processing including machine learning approaches.

Laser-induced fluorescence (LiF) provides a particularly well-suited alternative for REE identification. The distinct, narrow emission lines in LiF spectra correspond to element-specific electron excitation - radiative relaxation processes. The resulting energy release in form of photons shows characteristic wavelengths dependent on the electronic configuration of the elements (i.e. the trivalent REE with a partially filled 4f-shell), and on the element-neighbour configuration within the

crystal lattice (Gaft et al., 2005; Gaft and Panczer, 2013). The diagnostic LiF signatures deliver the spectral fingerprints that can be used in numerous applications in raw material characterisation.

Kauppinen et al. (2014) demonstrated the application of LiF for fast mineral mapping of e.g. drill cores, while others investigated the LiF for horizon control of mining machines (e.g., Nienhaus and Bayer, 2003), or quality control of mineral processing and sorting (e.g., Broicher, 2000, 2005). However, the mapping application cases focussed on general

measures of emission intensities or their ratios rather than on specific features in LiF spectra. Other studies emphasise on the high potential of LiF for REE detection in natural rocks (Reisfeld et al., 1996; Gaft et al., 2005; Lorenz et al., 2019; Seidel et al., 2019). Despite the promising studies, the method is not yet deployed in the raw material sector. To facilitate applications of LiF for REE detection in mineral exploration and related fields of material characterisation, the access to robust, traceable spectral reference data is needed.

LiF reference data is often very specific and limited (e.g. restricted to only few REE) and spectra are usually available only as plots (e.g., Czaja et al., 2008; Friis et al., 2010). Comprehensive summaries are given in form of tables with emission band positions along with information on excitation conditions (e.g., Gaft et al., 2005) and, similarly, in some online





databases (e.g. CSIRO, 2019; Barmarin, accessed: 2020-04-25). Such data limits emerging automated data analysis approaches to few parameters while other information stored in LiF spectra remains unspecified (e.g. relative intensities,

emission sub-levels). Although not within scope of the presented study, such information opens manifold possibilities also for future research questions not only in the mineral exploration field. Nevertheless, the availabilities of new, sophisticated, automated data processing routines emphasise already today the need for digital reference data of complete spectra, comparable to those for HSI, in order to further develop the LiF-based REE detection and analytical capacities for analyses of REE abundances and their spectral representation in natural rocks.

In this study, we present an open, digital LiF spectral library from REE phosphates as a traceable reference data repository that is available at zenodo.org, DOI: 10.5281/zenodo.4054606 (Fuchs et al., 2020). The library comprises the spectral information as high-resolution data in the visible to near-infrared range (350 nm - 1080 nm) to cover the major REE diagnostic emissions. LiF spectral fingerprints of REE in phosphates are important, because phosphates represent a group of typical REE-bearing minerals, e.g. monazite-(Ce), -(La), -(Nd) and -(Sm) (LREE($PO_4$)), xenotime-(Y) and -(Yb)

(Y($PO_4$) and Yb($PO_4$), respectively), or apatites ($Ca_5(PO_4)_3(OH,F,Cl)$). Those minerals are relevant in many REE deposits such as in Siilinjärvi and Korsnäs (Finland) (e.g., Goodenough et al., 2016), in Lovozero and Khibina (Kola Peninsula, Russia) (e.g., Kalashnikov et al., 2016) , in Bayan Obo (China) and Mt. Weld (Western Australia) (e.g., Emsbo et al., 2015; Lima and Filho, 2015) or in Ilimaussaq (South Greenland) (e.g., Zirner et al., 2015). We use reference material for electron microprobe analyses from the Smithsonian's Department of Mineral Sciences to base spectral results on

well-studied and referenced material (Jarosewich and Boatner, 1991; Donovan et al., 2002).

The scope of our study focuses on (1) characterising the observed emissions in LiF spectra of the Smithsonian REE phosphate standards by LiF, (2) investigating the suitability of three standard laser wavelengths (325 nm, 442 nm, 532 nm) with respect to excitation efficiency and selectivity, (3) introducing a LiF spectral library from REE phosphates as a transparent reference for robust evaluation of LiF data acquired from natural rocks, and (4) encouraging others to publish

spectral data to complement (a) the REE spectral library and (b) to expand the published knowledge on variations related to the chemical and mineralogical composition of natural (host) materials.

## 2 Material and methods

### 2.1 Rare-earth element standards

We used two sets of homogeneous, well-characterised, synthetic REE orthophosphates (crystal size of 0.5 - 3.0 mm)

from the Smithsonian National Museum of Natural History, Department of Mineral Sciences (sample IDs: NMNH 16484 - NMNH 168499, Jarosewich and Boatner 1991; Donovan et al. 2002, 2003, see Tab.1, for available data sheets see Smithsonian National Museum of Natural History 2019) to determine representative REE fluorescence spectra. The relatively simple crystallographic structure of the orthophosphates responsible for a simple defect configuration and the relatively pure chemical composition promote the material to deliver REE reference spectra without perturbances from

mixed REEs or complex mineral hosts of natural materials.




**Table 1.** Overview of rare earth element (REE) phosphate samples from the Smithsonian National Museum of Natural History used in this study for the LiF spectral library (based on Jarosewich and Boatner (1991); Donovan et al. (2002, 2003); LREE: light REE, HREE: heavy REE, ID: identification number of the Smithsonian National Museum of Natural History; data sheets see at: Smithsonian National Museum of Natural History (2019), Pb trace analysis values represent the average mass fraction of 10 measurements from instrumental neutron activation analysis, values are given in wt.-% $\times 10^{-2}$ to reduce number of zeros digits, given uncertainty values state one standard deviation).

| Sample | ID | $REE^{3+}$ (wt.-%) | $PO_4^{3-}$ (wt.-%) | Pb (wt.-% $\times 10^{-2}$) |
|---|---|---|---|---|
| **LREE** | | | | |
| $ScPO_4$ | NMNH 168495 | 32.12 | 67.88 | $0.00 \pm 0.00$ |
| $YPO_4$ | NMNH 168499 | 48.35 | 51.65 | $0.00 \pm 0.00$ |
| $LaPO_4$ | NMNH 168490 | 59.39 | 40.61 | $0.90 \pm 0.32$ |
| $CePO_4$ | NMNH 168484 | 59.60 | 40.40 | $1.90 \pm 0.07$ |
| $PrPO_4$ | NMNH 168493 | 59.73 | 40.27 | $0.92 \pm 0.04$ |
| $NdPO_4$ | NMNH 168492 | 60.30 | 39.70 | $0.86 \pm 0.17$ |
| $SmPO_4$ | NMNH 168494 | 61.28 | 38.72 | $0.86 \pm 0.13$ |
| $EuPO_4$ | NMNH 168487 | 61.54 | 38.46 | $0.64 \pm 0.16$ |
| $GdPO_4$ | NMNH 168488 | 62.34 | 37.66 | $0.39 \pm 0.16$ |
| **HREE** | | | | |
| $TbPO_4$ | NMNH 168496 | 62.59 | 37.41 | $0.00 \pm 0.00$ |
| $DyPO_4$ | NMNH 168485 | 63.11 | 36.89 | $0.00 \pm 0.00$ |
| $HoPO_4$ | NMNH 168489 | 63.45 | 36.55 | $0.00 \pm 0.00$ |
| $ErPO_4$ | NMNH 168486 | 63.78 | 36.22 | $0.00 \pm 0.00$ |
| $TmPO_4$ | NMNH 168497 | 64.01 | 35.99 | $0.00 \pm 0.00$ |
| $YbPO_4$ | NMNH 168498 | 64.56 | 35.44 | $0.00 \pm 0.00$ |
| $LuPO_4$ | NMNH 168491 | 64.81 | 35.19 | $0.00 \pm 0.00$ |

Previous analyses of the REE orthophosphates confirmed a high quality of the material regarding REE purity (Jarosewich and Boatner, 1991; Donovan et al., 2002, 2003). However, Donovan et al. (2002) detected REE cross-contamination of mass fractions at the order of $10^{-4}$ by instrumental neutron activation analysis (NAA) and found Pb contaminations in at least 7 out of the 16 samples (see Tab.1). Based on their analytical data, Donovan et al. (2002) determined a roughly 2 -
4% relative deviation from the expected (theoretical) sample composition, but found only the Pb concentration in $CePO_4$, $LaPO_4$ and $SmPO_4$ to possibly affect electron microprobe results. However, our recorded luminescence spectra do not indicate any Pb-related emission features.

Both sets belong to the same original population of REE reference samples for electron microprobe analysis (Jarosewich and Boatner, 1991) and differ only by one being embedded in a disc of epoxy resin and one being available in form of
single grains (see Fig.1). Together, the measured REEs in the given orthophosphate samples comprise Sc, Y, Pr, Nd, Sm, Eu, Tb, Dy, Ho, Er, Tm and Yb for embedded standards. From the same material production series, we additionally measured single grain specimen of the already mentioned REEs, plus Ce, La and Lu. No diagnostic emission lines are





expected for Sc, Y, La and Lu according to their electronic configuration and hence, those four REE were not included in our library. The diagnostic emission lines of Gd lie outside the detection range.

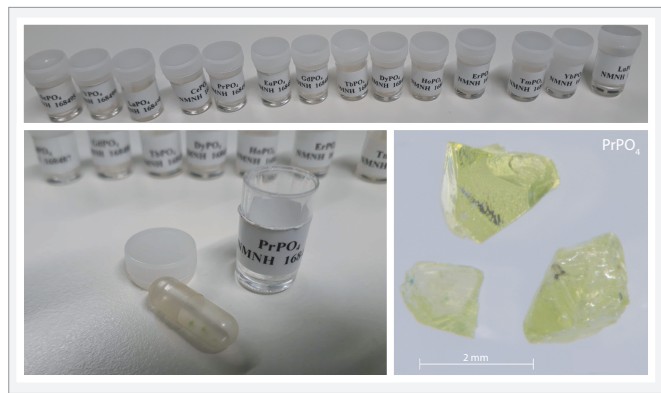

**Figure 1.** The reference REE orthophosphates from the National Museum of Natural History, example $PrPO_4$

## 2.2 Measurement setup

The measurement setup is shown in Fig.2. We used different laser wavelengths for LiF signal excitation: 325 nm (UV, He-Cd laser, focus spot 185 $\mu$m, 14.58 W/cm$^2$), 442 nm (blue, He-Cd laser, focus spot 170 $\mu$m, 16.73 W/cm$^2$) and 532 nm (green, diode-pumped frequency-doubled Nd:YAG, focus spot 143 $\mu$m, 26.93 W/cm$^2$). The focus spot sizes and power densities (given in previous sentence in brackets) are given as 1/e$^2$ of a Gaussian fit to the lateral intensity profile.

The emitted luminescence was dispersed using a 50 cm monochromator unit Acton SP2560i with a 300 gr/mm grating and recorded with a CCD camera Princeton Instruments SPEC-10:100BR_eXcelon (1340 channels, liquid nitrogen cooled). The used spectral detection range covers 340 nm to 1080 nm. Appropriate long pass filters (edges at 334 nm, 450 nm and 550 nm, respectively) ensured reliable separation of laser stimulation light from emitted luminescence, and were furthermore applied to suppress 2nd order diffraction signals in the long-wavelength part of the spectra. We measured all REE standards under continuous-wave excitation at room temperature and chose integration times between 300 ms and 10 s to optimise absolute signal intensities without saturating the CCD camera.

## 2.3 Data processing

The raw spectra were corrected for the spectral sensitivity of the detection unit with in-house calibration data for the respective technical setup. Consecutive data processing and visualisation were done using the R environment (R Core Team, 2014). Spectra over the full detection range of 340 - 1080 nm required two separate measurements (long-pass filter of e.g. 334 nm and 550 nm) to suppress the second order signals at double wavelength. For merging both, a merge-segment was defined with the starting point set to 560 nm and the upper end defined according to the minimum intensity



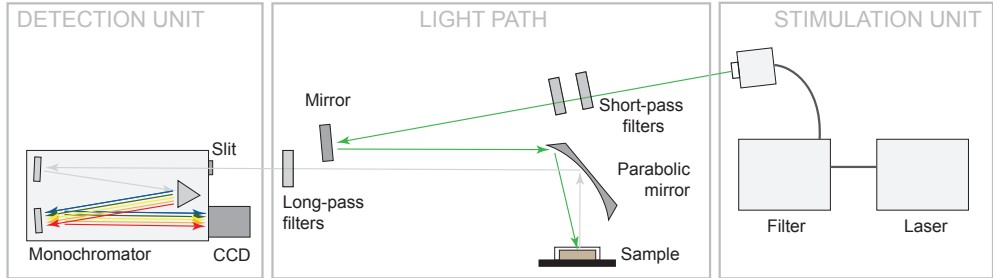

**Figure 2.** Scheme of technical setup for the laser-induced fluorescence (LiF) measurements

difference between both spectra. Within the merge-segment, both spectra are then merged using the intensity difference between both spectra multiplied by a weighting factor that gradually changes from 0 to 1. At the beginning of the merge-

segment, the difference multiplied by 0 is added to the first spectrum (measurements with 334 nm long-pass filter covering the short wavelength range), and then added after multiplying by the gradually increasing weighting factor until the end of the segment to become congruent with the second spectrum (measurements with 550 nm long-pass filter covering the longer wavelength range). Finally, the partial spectra were joined using the first spectrum at wavelengths shorter than 560 nm, the calculated new merge-segment of the overlapping range and the upper spectrum of longer wavelength above

the calculated merge-segment.

Spectra obtained from the set of embedded phosphate standards showed an intense broad-band emission attributed to the epoxy resin, in particular because of partly small and translucent sample crystals that are not perfectly exposed at the resin surface. We measured the epoxy resin at three locations next to REE crystals. The epoxy resin spectra show reproducible luminescence indicating material homogeneity and accordingly, allow for subtraction from the REE sample

spectra. Minor variations can be attributed to signal noise, which was removed by calculating a mean spectrum from the three epoxy resin spectra. We then subtracted the epoxy resin-related emission from the REE sample spectra. This operation compares to the removal of a non-sample related background continuum, allowing for an unbiased detection of remaining REE-related emission lines as well as of absorption features.

Peak detection was challenged by clustered emission lines and by distinguishing small REE emission peaks from noise

artefacts. Therefore, merged spectra were smoothed in a running mean interval of 1.2 nm to reduce the noise, while keeping the spectral sampling interval. The smoothing interval was carefully chosen to retain emissions of low intensity and details of clustered narrow emission lines. In the case of emission bands with clusters of many narrow emission lines, an automated peak detection was complicated and resulted typically only in the detection of the most prominent peak. To identify also the individual emission lines within emission bands, we calculated a background continuum based on the

spline smoothed running minimum (degrees of freedom 50, bin size 1.2 nm). After removing this background continuum, we detected all peaks above a threshold of typically two standard deviations of the entire spectrum.





### 2.4 Sample homogeneity

Recorded LiF spectra represent a single measurement spot, where the signal to noise ratio was best. We checked for spatial variations and representativity of the selected LiF measurement spot using hyperspectral reflectance imaging. The images from an FX10 camera (Spectral Imaging Ltd.) with a spatial resolution of 0.17 mm depict the sample grains with 5 to 25 pixels (pure sample pixels), each containing the reflectance spectrum in a range of 400 nm - 1000 nm. All spectra of the measured REE sample grain were analysed for features deviating from expected REE absorptions. The reflectance data was analysed using the R package 'hyperSpec' (Beleites and Sergo, 2018) and compared to the reflectance spectral library of the USGS (Kokaly et al., 2017).

The obtained reflectance spectra of pure sample pixels were consistent with respect to absorption feature position (Fig.3). Variations in signal intensities have minor influences on relative absorption depth (standard deviation below 5 %). The reference spectral library for standard material from USGS (Kokaly et al., 2017) comprises only REE oxides and in some cases chlorides, but no phosphates, but comparison of absorption position confirmed agreement with observed absorption features on our REE samples. None of the pure sample spectra contained additional absorption features and thus, reflectance spectra attest the spatial homogeneity and good sample quality.

## 3 Results

### 3.1 REE reference spectra at 325 nm laser excitation

With a laser excitation at 325 nm the Smithsonian REE phosphate samples yielded representative spectra for $Ce^{3+}$, $Nd^{3+}$, $Eu^{3+}$, $Tb^{3+}$, and $Dy^{3+}$ (Fig.4). Prominent emission lines with high signal to noise ratios allow for an unequivocal assignment of transitions, and represent the diagnostic features needed for a LiF-based REE identification. The high spectral resolution (spectral sampling 0.13 nm) of the measurement setup enables to separate multiple lines within emission clusters associated to splitting of transitions into sub-levels (Stark level splitting) as observed particularly in the spectra of $Nd^{3+}$, $Tb^{3+}$ and $Dy^{3+}$ (Fig.4). Table 2, therefore, differentiates between most intense emission lines (main) and additionally detected less intense emission lines of the same emission band (minor).

For $CePO_4$, the excitation results in a broadband emission between 374 nm and 540 nm (here given as FWHM: full width at half maximum without prior background removal as this is not required for such broadband emission) and a maximum at 420 nm (Fig.4). The very broad emission is difficult to interpret based on our results and goes beyond the scope of this study. In general, it overlaps with the characteristic $Ce^{3+}$ emission band around 360 nm that is associated to a 5d - 4f transition (e.g., Reisfeld et al., 1996; Gaft et al., 2005; Shalapska et al., 2014). Several studies report broadened Ce-related peaks and discuss reasons behind, e.g. Shalapska et al. (2014) identified a peak doublet with the peak distance dependent on the site symmetry in the crystal lattice, and Chen et al. (2014) observed broadened emission in the range between 310 - 410 nm resulting from mixed phases of a monoclinic and hexagonal lattice. The Ce-related emission band appears less broad and shifts towards lower wavelengths in the two, itself non-luminescent samples of



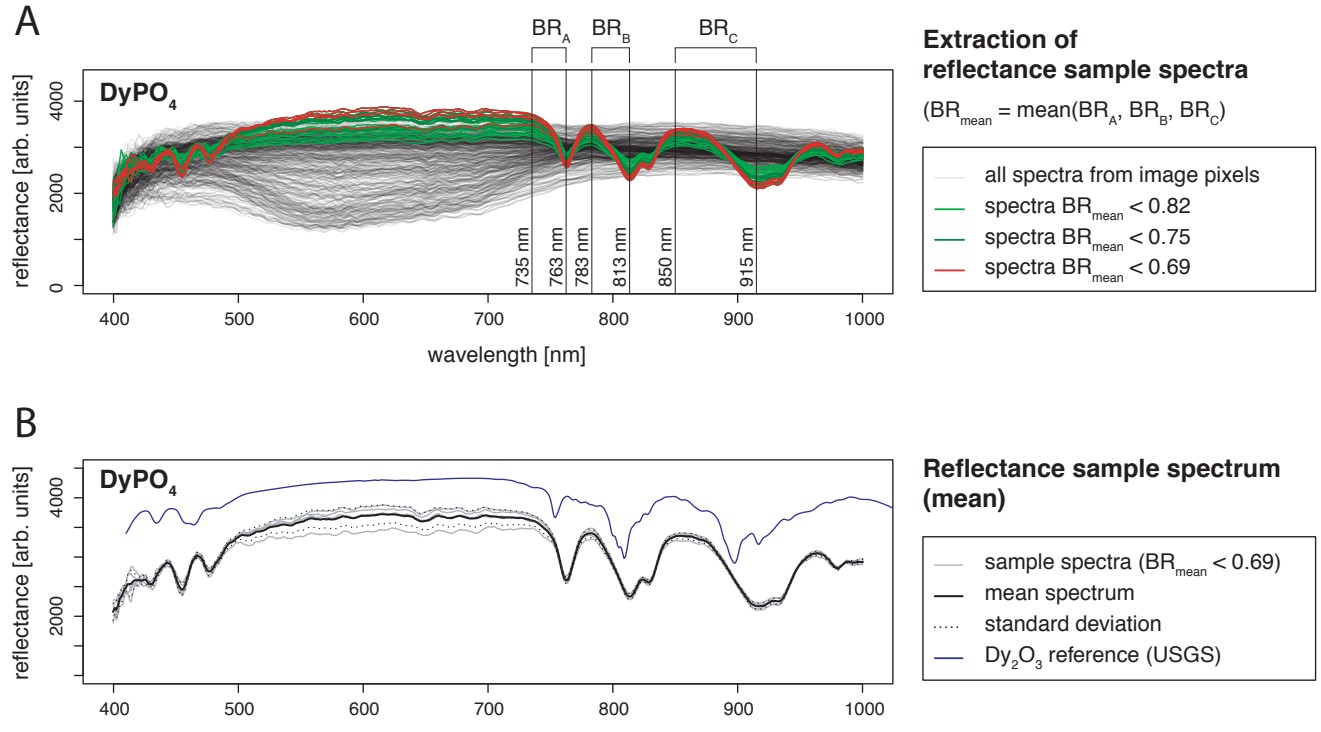

**Figure 3.** Spatial homogeneity evaluation with hyperspectral reflectance images from a SPECIM FX10 camera with a spatial resolution of 0.17 mm, example $DyPO_4$. A: For REE spectra extraction from all spectra of a measured scene (image), the mean of three band ratios was calculated to reduce influences from noise and ensure accurate identification of sample spectra (pixels containing spectra from the sample grain, in the text referred to as pure sample pixels). Bands were selected according to position of diagnostic absorption features (BR = band ratio representing the ratio of signal intensities at two given wavelengths). B: A mean spectrum was calculated from all pure sample pixels. The standard deviation is given as an indication of the variability between spectra of pure sample pixels. The resulting REE absorption spectrum (mean spectrum) was compared to reference data from the USGS spectral library (Kokaly et al., 2017).

LuPO$_4$ and YPO$_4$ (grey lines in Fig.4, both samples from same set of Smithsonian REE phosphate standards, cf. Tab.1). LuPO$_4$ contains Cerium at a mass fraction of $(0.1 \pm 0.2) \times 10^{-3}$ (NAA results according to Donovan et al. 2002) and reveals the maximum emission of $Ce^{3+}$ at 362 nm (FWHM 356 - 373 nm). YPO$_4$ contains Cerium at a mass fraction of $(0.5 \pm 0.6) \times 10^{-3}$ (NAA results according to Donovan et al. 2002) and here the maximum emission of $Ce^{3+}$ lies at 359 nm (FWHM 352 - 365 nm).

       The NdPO$_4$ spectrum depicts a prominent cluster of distinct emission lines peaking at 1059 nm and a secondary cluster
of less intense emission at 908 nm (Fig.4). The two emission bands represent $^4F_{3/2}$ - $^4I_{11/2}$ and $^4F_{3/2}$ - $^4I_{9/2}$ transitions,

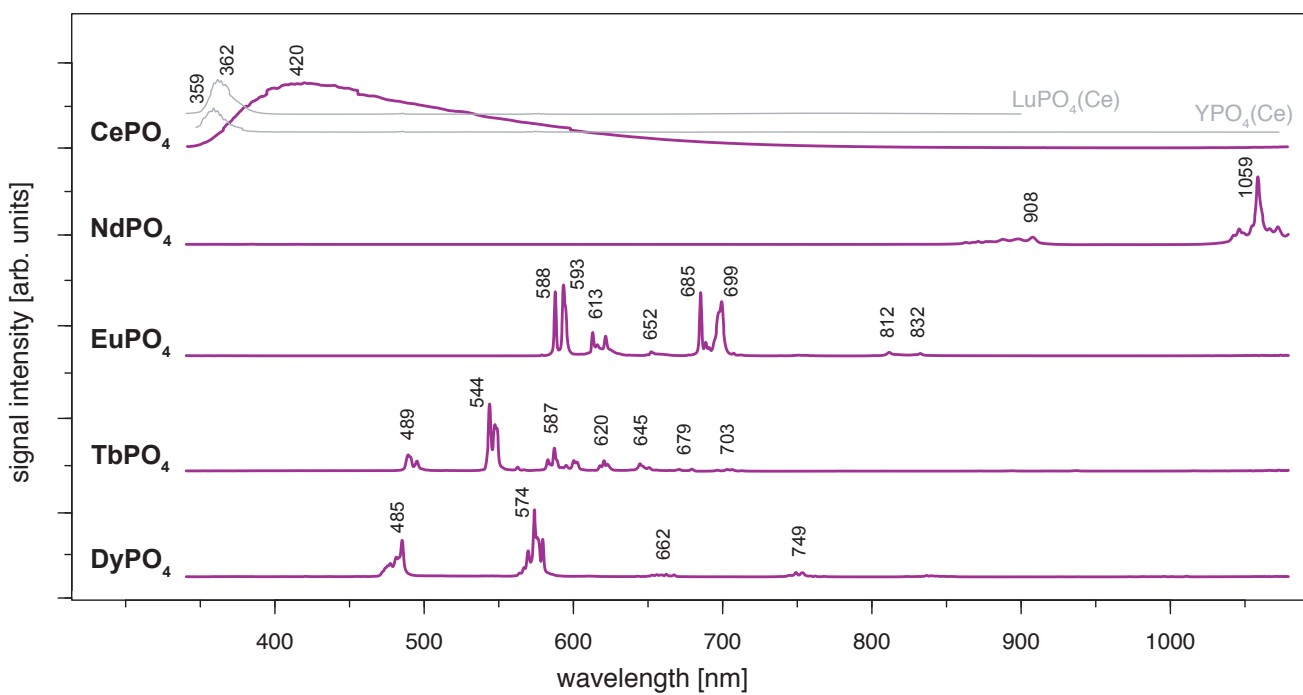

**Figure 4.** LiF reference spectra obtained with 325 nm laser excitation depict the diagnostic emission lines for the REE phosphates NdPO$_4$, EuPO$_4$, TbPO$_4$, and DyPO$_4$ (given values represent wavelength of most intense emission in nm for each transition, cf. Tab. 2). CePO$_4$ is included for completeness, diagnostic emissions are shown by the grey lines representing spectra from two non-luminescent REE PO$_4$ samples with Cerium contaminations (see further explanation in the text along with Cerium concentrations in those samples based on neutron activation analysis (NAA) results)

respectively (e.g. Reisfeld et al., 1996; Gaft et al., 2005; Czaja et al., 2012; Shalapska et al., 2014). Emission bands at shorter wavelengths (< 850 nm, Gaft et al. 2005) were not observed during measurements and hence, are not recorded in our phosphate reference spectra. Nevertheless, the two observed Nd$^{3+}$ emission bands are prominent and represent diagnostic features as they do not overlap with emissions of other REEs.

The spectrum of EuPO$_4$ (Fig.4) shows distinct emission lines at 588 nm, 593 nm, 612 nm, 651 nm and 699 nm, representing five transitions from $^5$D$_0$ to $^7$F$_i$ ($i$ = 0, 1, 2, 3, 4, Gaft et al. 2005; Friis et al. 2010; Shalapska et al. 2014). The spectrum indicates splitting into sub-level emissions. Triplets of emission lines are recorded for the transitions to $^7$F$_2$ and $^7$F$_4$ with additional prominent lines at 622 nm and 685 nm, and a line of minor intensity at 616 nm and 689 nm, respectively. We observe two additional weak emission lines at 812 nm and 832 nm. These rarely described emissions relate

to the $^7$F$_6$ transition (Binnemans, 2015). The emission pattern corresponds to Eu$^{3+}$ observed in apatite (Reisfeld et al., 1996), where the ratio of emission intensities for $^5$D$_0$ to $^7$F$_2$ and $^5$D$_0$ to $^7$F$_1$ is low. However, the detailed spectroscopic sample characterisation can be found in Sharma et al. (2019).



The emission lines in the $TbPO_4$ spectrum represent five prominent transitions (from $^5D_4$ to $^7F_i$, $i$ = 6, 5, 4, 3, 2) and two weak transitions (from $^5D_4$ to $^7F_1$ and $^7F_0$, (Nazarov et al., 2009; Shalapska et al., 2014; Fang et al., 2017). The transitions are evident as clusters of multiple emission lines with maximum intensities at 489 nm, 544 nm, 587 nm, 587 nm, 620 nm, 645 nm, 679 nm and 703 nm (Fig.4). Reisfeld et al. (1996) and Gaft et al. (2005) reported emission lines below 440 nm for the phosphate minerals apatite and monazite. Those transitions from $^5D_3$ to $^7F_i (i = 6, 5, 4)$ are not evident in our sample spectrum.

The $DyPO_4$ spectrum displays narrow emission lines with two prominent bands at 485 nm, 574 nm and two weak emission bands at 662 nm and 749 nm. The clustered emission lines correspond to the transitions $^4F_{9/2}$ to $^6H_i$ ($i$ = 15/2, 13/2, 11/2, 9/2, Reisfeld et al. 1996; Gaft et al. 2005; Friis et al. 2010). Our instrumental setup allows for a detection of separate sub-level emission lines corresponding to effects of the phosphate crystal field. Absorption spectroscopy suggests sufficient $Dy^{3+}$ excitation when using a 325 nm laser, although higher efficiency can potentially be achieved using slightly longer wavelength laser (e.g. 352 nm or 366 nm, Friis et al. 2010).

## 3.2 Absorption features in REE spectra at 325 nm laser excitation

Spectra of several REE phosphates, especially $PrPO_4$, $SmPO_4$, $HoPO_4$, $ErPO_4$ and $TmPO_4$ are dominated by a very broad emission band between 350 nm and 800 nm when using 325 nm laser excitation (Fig. 5), irrespective of the sample being embedded in epoxy resin (see Fig. 5 dark-violet spectra, D = epoxy disc sample) or in form of stand-alone single grains (see Fig. 5 violet spectra, SG = single grain sample). The broad emission band is intersected by prominent absorption features. Comparison to the USGS spectral library for reflectance data (Kokaly et al., 2017) and absorption positions reported in the literature (e.g., White, 1967; Boesche et al., 2015; Turner et al., 2014) confirms absorptions to match with respective REE-reference spectra.

The broad, unspecific emission band causes strong masking of most REE-diagnostic emission lines <650 nm. In case of REE with diagnostic emission lines >650 nm, the broad-band emission is less problematic because observed patterns in the long wavelength region may be sufficient for the REE identification (e.g. in the case of $ErPO_4$, spectrum of the SG: single grain specimen in Fig. 5). Such 325 nm-excited spectra depicting both, absorptions and emissions in the same spectrum. The REE-related absorption features are recorded at wavelength <800 nm and diagnostic emission lines in the spectral region >650 nm. To suppress the broad-band emission and unravel the REE emission lines of interest, longer wavelength laser excitation proved successful.

## 3.3 REE reference spectra at 442 nm laser excitation

When using 442 nm laser excitation, the same diagnostic emission patterns were recorded for the above presented REE $PO_4$ samples ($NdPO_4$, $EuPO_4$, $TbPO_4$, and $DyPO_4$). Only the excitation efficiency appeared to be less as signal to noise ratios decreased, in particular for $NdPO_4$ but also for $TbPO_4$, and $DyPO_4$. Additionally, representative LiF spectra could be acquired for $PrPO_4$, $SmPO_4$ and $ErPO_4$.





**Table 2.** Overview of detected emission lines and corresponding transitions for the REE phosphates $NdPO_4$, $EuPO_4$, $TbPO_4$, and $DyPO_4$ that were successfully excited at 325 nm and Cerium emission measured in $YPO_4$ and $LuPO_4$ ($\lambda_{exc}$: excitation wavelength, $\lambda_{emi}$: emission wavelength, main: most intense emission line, minor: additionally detected, less intense emission lines of the same emission band)

| $\lambda_{exc}$: 325 nm | | | |
|---|---|---|---|
| **Center** | $\lambda_{emi}$ **(main)** | $\lambda_{emi}$ **(minor)** | **Transition** |
| | (nm) | (nm) | |
| $Ce^{3+}$ [a] | 359 | | 5d - 4f |
| $Ce^{3+}$ [b] | 362 | | 5d - 4f |
| $Nd^{3+}$ | 908 | 863, 888, 898 | $^4F_{3/2}$ - $^4I_{9/2}$ |
| | 1059 | 1043, 1046, 1067, 1072 | $^4F_{3/2}$ - $^4I_{11/2}$ |
| $Eu^{3+}$ | 588 | | $^5D_0$ - $^7F_0$ |
| | 593 | | $^5D_0$ - $^7F_1$ |
| | 613 | 616, 622 | $^5D_0$ - $^7F_2$ |
| | 652 | | $^5D_0$ - $^7F_3$ |
| | 685 | 689, 699 | $^5D_0$ - $^7F_4$ |
| | | 812, 832 | $^5D_0$ - $^7F_6$ |
| $Tb^{3+}$ | 489 | 495 | $^5D_4$ - $^7F_6$ |
| | 544 | 547, 549 | $^5D_4$ - $^7F_5$ |
| | 587 | 582, 595, 600, 602 | $^5D_4$ - $^7F_4$ |
| | 620 | 618, 622 | $^5D_4$ - $^7F_3$ |
| | 645 | 650 | $^5D_4$ - $^7F_2$ |
| | 679 | 671 | $^5D_4$ - $^7F_1$ |
| | 703 | 706 | $^5D_4$ - $^7F_0$ |
| $Dy^{3+}$ | 485 | 477, 481 | $^4F_{9/2}$ - $^6H_{15/2}$ |
| | 574 | 570, 579 | $^4F_{9/2}$ - $^6H_{13/2}$ |
| | 662 | 656, 667 | $^4F_{9/2}$ - $^6H_{11/2}$ |
| | 749 | 753 | $^4F_{9/2}$ - $^6H_{9/2}$ |

[a] measured in $YPO_4$ (Ce at mass fraction of $(0.1 \pm 0.2) \times 10^{-3}$, Donovan et al. 2002)

[b] measured in $LuPO_4$ (Ce at mass fraction of $(0.5 \pm 0.6) \times 10^{-3}$, Donovan et al. 2002)





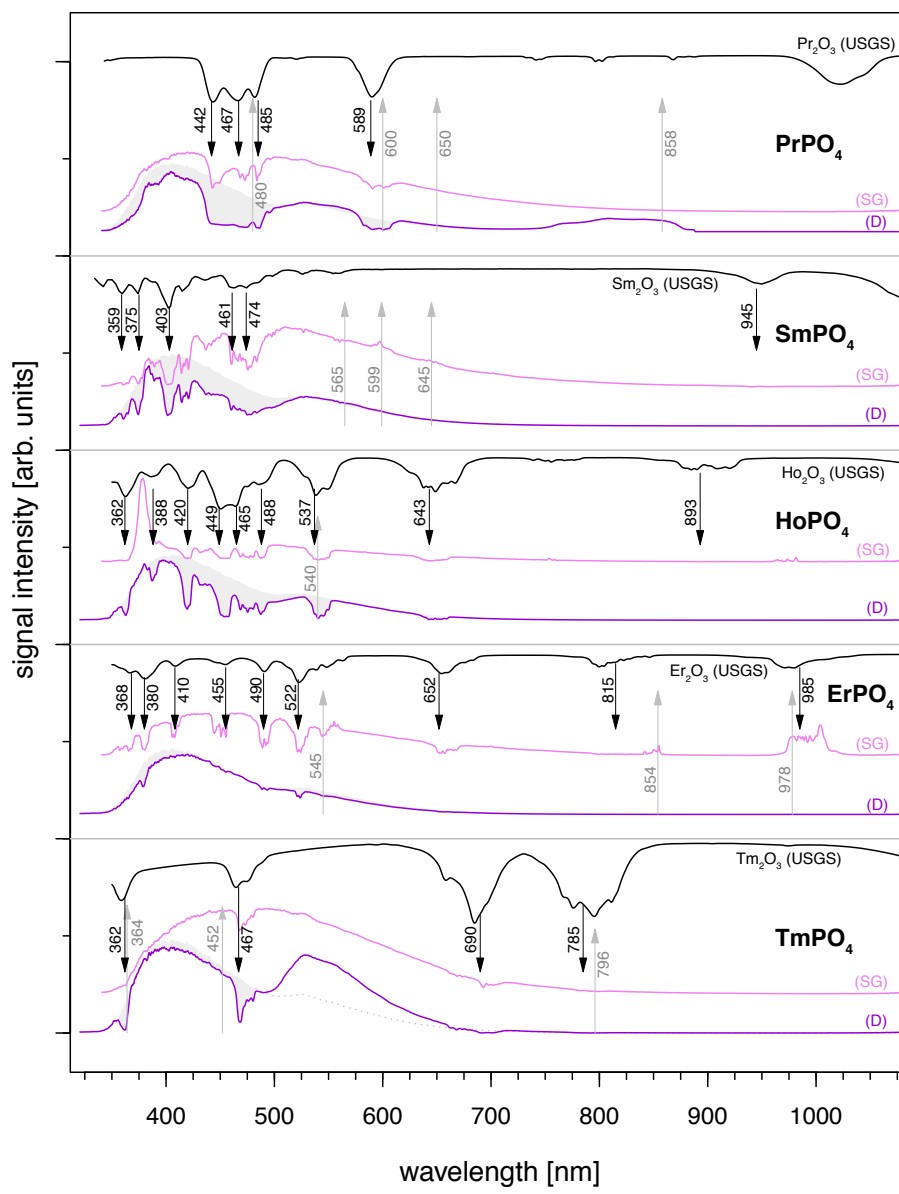

**Figure 5.** REE spectra illustrating diagnostic absorption features within prominent broad-band emission at wavelengths <800 nm and REE emission lines for those >650 nm. Violet and pink PL spectra were measured using laser excitation at 325 nm for both sample specimen (D: samples embedded in disc, SG: single grain samples, grey shaded area represents epoxy resin mean spectrum expected for D samples without absorption features, emissions >650 nm are better visible in SG samples due to the lower intensity of the broad-band luminescence). Black reflectance spectra depict reference data from the USGS library (Kokaly et al., 2017)





Excitation with the 442 nm laser allows to unravel distinct $Pr^{3+}$-related emission lines in the $PrPO_4$ reference spec-
       trum at 498 nm, 526 nm, 617 nm, 642 nm and 724 nm (see Fig. 6 and Tab. 3) corresponding to the dominant transitions
       described by Gaft et al. (2005) (first, third and fourth emission line) and Zhou et al. (2012) (first four emission lines). Ad-
       ditionally, Liang et al. (2017), Prasad et al. (2018) and Runowski et al. (2019) recorded also the emission above 700 nm.
       The transitions originate in $^3P_0$ or $^3P_1$ (Zhou et al., 2012; Prasad et al., 2018; Runowski et al., 2019), where the com-
parison between reported and observed emission wavelength indicates that both origins are possible and the (relative)
       dominance of one or the other decided about the observed emission position (see Tab. 3). The transition's final states are
       $^3H_i$ ($i$ = 4, 5, 6) and $^3F_j$ ($j$=2, 4) (Zhou et al., 2012; Shalapska et al., 2014; Prasad et al., 2018; Runowski et al., 2019).
       Prasad et al. (2018) identified the $^1D_2$ state to be the transition origin for the prominent Pr emission at around 600 nm
       as well as Gaft et al. (2005), who also attributed the emission around 500 nm to the same origin. However, the observed
positions lie at slightly (around 10 nm) higher emission wavelength that agree better to positions found by e.g. Friis et al.
       (2010) and Runowski et al. (2019), and potentially result from a superposition of different transitions originating to certain
       portions from $^3P_0$, $^3P_1$ and $^1D_2$. Nevertheless, the origin of this rather large shift of 10 nm is currently unknown and will
       require in-depth investigation beyond the scope of this study.

       The 442 nm laser-induced $SmPO_4$ spectrum depicts prominent emission line clusters at 560 nm, 597 nm, 643 nm and
a cluster of minor intensity at 702 nm (Fig. 6). The first three emission lines correspond to $Sm^{3+}$ transitions from $^4G_{5/2}$
       to $^6H_i$ ($i$ = 5/2, 7/2, 9/2, see Tab. 3) in line with observations of Gaft et al. (2005), Friis et al. (2010), Czaja et al. (2012),
       Shalapska et al. (2014) and Guan et al. (2016). The pattern of four emission lines with the second line as most efficient
       emission resembles well observations by (e.g. Ha et al., 2016), Wantana et al. (2017) or Yashodha et al. (2019), who
       assigned the fourth transition to $^6H_{11/2}$. However, this transition has been attributed to a luminescence band observed at
720 nm by Friis et al. (2010). Alternatively, the fourth observed emission line also coincides with the position described
       for $Sm^{2+}$ centres and respective transitions from $^7D_0$ to $^7F_i$ ($i$ = 1, and potentially 0 and 2) (Gaft et al., 2005). Their
       emission overlap may in turn explain the broadened but weak emission cluster at around 700 nm. Another explanation
       for the observed 700 nm emission cluster and potential source for peak broadening of the around 600 nm and 700 nm
       emission clusters could be the small detected Eu contamination in the $SmPO_4$ sample (see NAA results in Donovan et al.
2002). The very weak near infrared emission features around 900 - 950 nm may indicate additional, less often observed
       Sm emission features from $^4G_{5/2}$ to $^6F_i$ transitions ($i$ = 1/2, 3/2, 5/2) (e.g., Ahmed and Iftikhar, 2019), where e.g. Samanta
       et al. (2016) ascribe the latter transition around 950 nm to energy transfer from Ce (see small contaminations of Ce in
       NAA results in Donovan et al. 2002). A Nd contamination with corresponding emission lines in the same wavelength
       region (around 900 nm) is not evident in NAA results (Donovan et al., 2002).

Laser excitation of $ErPO_4$ at 442 nm unravels an emission at 550 nm, besides the emission at 855 nm and the prominent
       cluster between 977 nm and 1004 nm (see Fig. 6), which were already detected at 325 nm excitation (see above). This
       first line represents an important diagnostic emission in the visible range (e.g., Reisfeld et al., 1996; Gaft et al., 1998; Friis
       et al., 2010). Gaft et al. (2005) assign respective emission lines to transitions from $^4S_{3/2}$ to $^4I_i$ ($i$ = 15/2, 9/2) (first two
       lines) and $^4I_{11/2}$ to $^4I_{15/2}$ (third line) (see Tab. 3). Among others, Czaja et al. (2012) attribute another emission line at

 

**Table 3.** Overview of detected emission lines and corresponding transitions for those REE that were successfully excited with the 442 nm laser (potential assignments to transitions are given in brackets)

| $\lambda_{exc}$: 442 nm | | | |
|---|---|---|---|
| **Center** | $\lambda_{emi}$ **(main)** (nm) | $\lambda_{emi}$ **(minor)** (nm) | **Transition** |
| $Pr^{3+}$ | 498 | | $^3P_0$ - $^3H_4$ |
| | 526 | 539, 544, 555 | $^3P_0$ - $^3H_5$ |
| | 617 | 608, 625 | $(^3P_1, ^3P_0)$ - $^3H_6$ |
| | 642 | 637, 647 | $^3P_0$ - $^3F_2$ |
| | 724 | | $^3P_0$ - $^3F_4$ |
| $Sm^{3+}$ | 560 | 567 | $^4G_{5/2}$ - $^6H_{5/2}$ |
| | 597 | 593, 607 | $^4G_{5/2}$ - $^6H_{7/2}$ |
| | 643 | 637, 649, 654 | $^4G_{5/2}$ - $^6H_{9/2}$ |
| | 702 | 716 | $(^4G_{5/2}$ - $^6H_{11/2})$ |
| | 900 - 950 | | $^4G_{5/2}$ - $^6F_{(1/2,3/2,5/2)}$ |
| $(Sm^{2+}$ | 702 | 716 | $^7D_0$ - $^7F_1)$ |
| $Er^{3+}$ | 550 | 547 | $^4S_{3/2}$ - $^4I_{15/2}$ |
| | 669 | | $^4F_{9/2}$ - $^4I_{15/2}$ |
| | 855 | 842, 850 | $^4S_{3/2}$ - $^4I_{9/2}$ |
| | 1004 | 977, 985, 990, 994 | $^4I_{11/2}$ - $^4I_{15/2}$ |

about 653 nm to $Er^{3+}$ that is only weakly present in our reference $ErPO_4$ spectrum (see Fig. 6) and identify the transition from $^4F_{9/2}$ to $^4I_{15/2}$ to be responsible for this emission.

### 3.4 REE reference spectra at 532 nm laser excitation

For $HoPO_4$, the diagnostic emission line around 540 nm (e.g., Gaft et al., 2005; Qin et al., 2011; Pandey and Swart, 2016) was masked by broad-band luminescence with $Ho^{3+}$-related absorptions (e.g., Turner et al., 2014; Kokaly et al., 2017),
when using 325 nm and 442 nm laser excitation (see above). The 532 nm excited spectrum captures only emissions above 550 nm, but successfully suppressed the masking broad-band luminescence below 700 nm. Recorded emission lines cluster at 662 nm, 754 nm and 982 nm (see Fig. 7 and Tab. 4). The first, relatively weak emission line corresponds to a $^5F_5$ to $^5I_8$ transition (e.g., Friis et al., 2010; Qin et al., 2011; Pandey and Swart, 2016), while the second, more intense emission cluster results from a $^5F_4/^5S_2$ to $^5I_7$ transition (e.g., Qin et al., 2011; Pandey and Swart, 2016). The third and
most prominent emission cluster in the near-infrared wavelength range resembles observations by Yu et al. (2012), who detect a transition from $^5F_5$ to $^5I_7$ with emission lines at 965 nm. A combination with transition processes assigned to

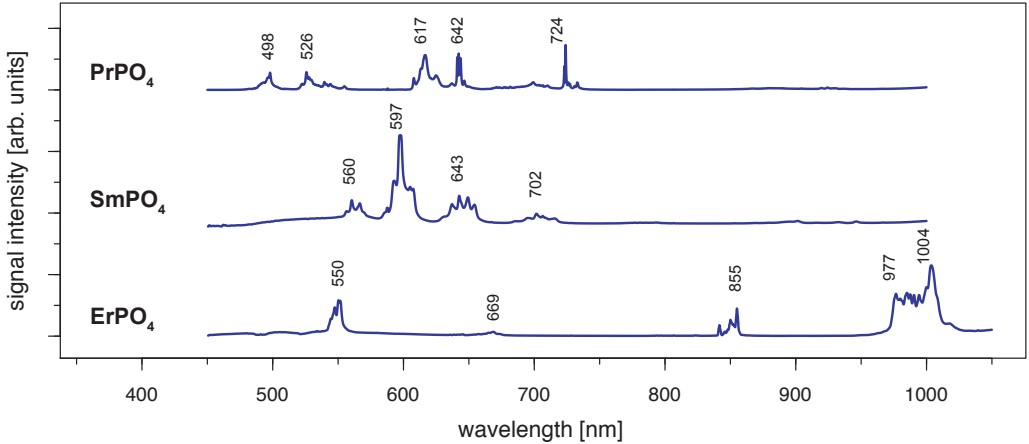

**Figure 6.** LiF reference spectra obtained with 442 nm laser excitation depict the diagnostic emission lines for the REE phosphates $PrPO_4$, $SmPO_4$ and $ErPO_4$ (given values represent wavelength of most intense emission in nm for each transition, cf. Tab. 3)

a $^5F_4$/$^5S_2$ to $^5I_6$ with typical emission features at longer wavelengths conjecturally explain the observed broad, intense emission cluster at 982 nm. An influence from $Yb^{3+}$ as intensifier of this spectral response (Yu et al., 2012) can be excluded by NAA results below detection limit (Donovan et al., 2002).

Depending on the embedding material of measured $NdPO_4$ grains, the laser excitation with 325 nm may cause a broad-band luminescence below 700 nm, while the use of the 442 nm laser revealed inefficient $Nd^{3+}$ excitation. Instead, the 532 nm excitation proved efficient and revealed two major emission line clusters, one between 860 nm and 910 nm (maximum at 908 nm) and the other one from 1040 nm to 1075 nm (maximum at 1059 nm) (see Fig. 7). The two emission clusters agree to the $Nd^{3+}$-related lines resulting from a $^4F_{3/2}$ to $^4I_{9/2}$ and a $^4F_{3/2}$ to $^4I_{11/2}$ transition, respectively, and

resemble common observations in phosphate minerals at 532 nm laser excitation (Tab. 4) (e.g., Reisfeld et al., 1996; Gaft et al., 2005; Czaja et al., 2012; Shalapska et al., 2014). Reported emission lines below 800 nm by Gaft et al. (2005) are not present in any of our reference $NdPO_4$ spectra. Shalapska et al. (2014) identified additional emissions to be too weak, of which, however, the one to $^4I_{15/2}$ at 846 nm (cf. Gaft et al. (2005)) may be also present in our reference spectrum contributing to the broad cluster around 908 nm.

The 532 nm laser excitation of $YbPO_4$ revealed a relatively weak $Yb^{3+}$ emission without the strong broad-band emission at shorter wavelength than 700 nm observed in several of our REE $PO_4$ samples at 325 nm excitation (see Sect. 3.2). The diagnostic $Yb^{3+}$ emission appears as a relatively broad band with a maximum at 1011 nm and a tentative shoulder at 1023 nm (Fig. 7). Results correspond to the main emission line reported by Gaft et al. (2005) and Czaja et al. (2012) and reflect a ($^2F_{5/2}$, $^4F_{5/2}$) to $^2F_{7/2}$ recombination process (Tab. 4). Potentially overlapping emissions from splitting into Stark

levels (Gaft et al., 1998, 2005; Czaja et al., 2012) may explain the broadened emission band with peak shoulders. The





**Table 4.** Overview of detected emission lines and corresponding transitions for those REE that were successfully excited with the 532 nm laser

| $\lambda_{exc}$: 532 nm | | | |
|---|---|---|---|
| **Center** | $\lambda_{emi}$ **(main)** | $\lambda_{emi}$ **(minor)** | **Transition** |
| | (nm) | (nm) | |
| $Ho^{3+}$ | 662 | | $^5F_5$ - $^5I_8$ |
| | 754 | 743, 747, 759 | $^5F_4$/$^5S_2$ - $^5I_7$ |
| | 982 | 964, 968, 974, 976 | $^5F_5$ - $^5I_7$ |
| $Nd^{3+}$ | 908 | 899 | $^4F_{3/2}$ - $^4I_{9/2}$ |
| | 1059 | 1046, 1067, 1072 | $^4F_{3/2}$ - $^4I_{11/2}$ |
| $Yb^{3+}$ | 1011 | 1023 | $(^2F_{5/2}, {}^4F_{5/2})$ - $^2F_{7/2}$ |

interpretation of overlapping emission lines agrees also with observations by Liang et al. (2016) who detected intense photoluminescence with main peaks at 980 nm and 1019 nm and several weak peaks at 1020 - 1100 nm.

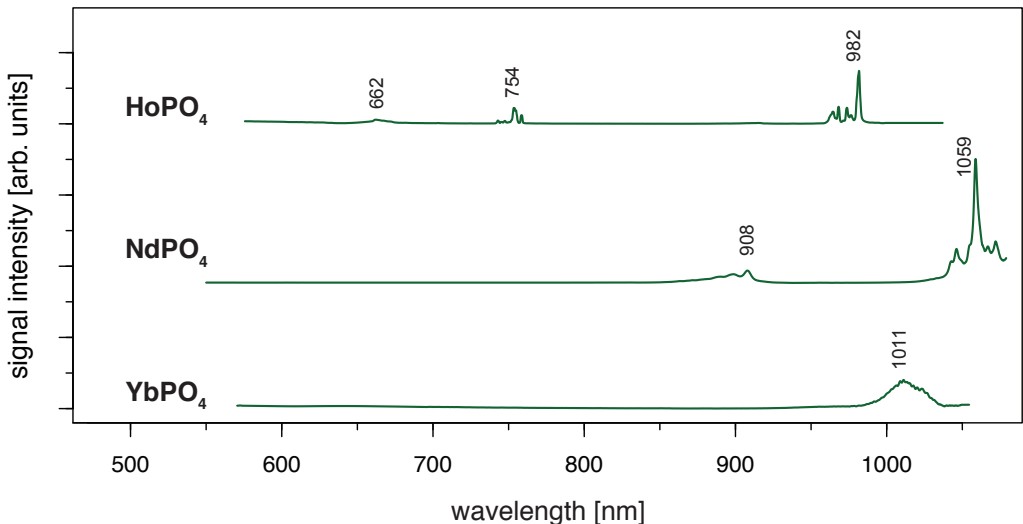

**Figure 7.** LiF reference spectra obtained with 532 nm laser excitation depict the diagnostic emission lines for the REE phosphates HoPO$_4$, NdPO$_4$ and YbPO$_4$ (given values represent wavelength of most intense emission in nm for each transition, cf. Tab. 4)



## 4 Discussion

### 4.1 Suitable REE excitation wavelength

The REE phosphate reference spectra presented above show that adequate excitation conditions can be achieved when selecting one out of the three laser wavelength used in this study. Limiting suitable conditions for all REE to three excitation wavelengths delivers great advantages, when looking for technical implications for e.g. automated, cost-efficient sensor solutions. The results indicate sufficient conditions for recording the main diagnostic emission lines and hence, for identifying the respective REE (with the exception of $GdPO_4$ and $TmPO_4$), although the used wavelengths do not

perfectly match with the most efficient excitation wavelength of individual REE.

The efficiency of excitation can be evaluated based on absorption spectroscopy, where prominent absorption features indicate energy uptake by the crystal. Figure 8 depicts the three laser wavelengths (325 nm, 442 nm, 532 nm) compared to our own absorption spectra using the same REE phosphate samples from the Smithsonian library measured with the portable field-spectrometer Spectral Evolution PSR 3500 (Fig. 8, orange line). Complementary, Figure 8 shows re-

flectance spectra from synthetic REE phosphates presented by Ropp (1969) (black line), and reference data from REE oxides available from Kokaly et al. (2017) (grey line). We included the REE oxides as an additional reference mainly because of the wide use of this spectral library and the overall good match of absorption positions.

The laser wavelength of 325 nm does match with positions of absorption features for $Tb^{3+}$ and $Dy^{3+}$ or coincide with shoulders of absorption features for $Ce^{3+}$, $Eu^{3+}$, and potentially also for $Nd^{3+}$. This superposition indicates good exci-

tation conditions. For the other REEs, the 325 nm laser line lies outside or between absorption features suggesting less suitable conditions. Blue laser excitation at 442 nm is most efficient for $Pr^{3+}$, matches also for $Sm^{3+}$ and $Er^{3+}$, and overlaps with an absorption shoulder in $Dy^{3+}$ and $Ho^{3+}$. The 532 nm laser excitation showed successful in particular for $Nd^{3+}$ and also $Ho^{3+}$. The good excitation conditions with green laser light for $Nd^{3+}$ agree to other studies on excitation wavelength (e.g. Czaja et al., 2012; Lenz, 2015), while excitation at 325 nm was concluded to be inefficient. Fig. 8 suggests

that a 325 nm laser is also able to excite Nd according to the coincidence with the edge range of an absorption feature. Nevertheless, the real configuration of the crystal lattice apparently influences intensity ratios of sub-level emissions, and shifting emission band positions (Lenz et al., 2013). For $Yb^{3+}$, it seems surprising that 532 nm as well as 325 nm excitation were successful, because both excitation wavelengths lie at longer wavelengths than the absorption at $< 250$ nm shown in Ropp (1969) (cf. Fig. 8). However, for example Liang et al. (2016) or Chakraborty et al. (2016) found also

efficient absorption of excitation energy between 300 nm and 400 nm to charge transfer states that emit at wavelengths longer than 900 nm. Another possible explanation may be the sensitisation of $Yb^{3+}$ by traces of other REE detected in the studied material by NAA (Donovan et al., 2002). The trace contaminations result from the reference material's production procedure, where the chemical similarity of REEs limits the complete separation of REE in the samples. Gaft et al. (2005) report $Yb^{3+}$ excitation to be strongly dependent on Nd, but NAA results from Donovan et al. (2002) do not indicate any Nd

contamination. Other potential candidates for energy transfer processes to $Yb^{3+}$ are Ho (e.g., Wei et al., 2011; Gavrilović et al., 2015; Kang et al., 2019) (NAA mass fraction: $0.2 \pm 0.4 \times 10^{-3}$), Er (e.g., He et al., 2017; Wang et al., 2018; Kang





et al., 2019) (NAA mass fraction: $0.2 \pm 0.2 \times 10^{-3}$), Tb (e.g., Gandhi et al., 2014) (NAA mass fraction: $0.2 \pm 0.3 \times 10^{-3}$), and Ce (e.g., Tang et al., 2018) (NAA mass fraction: $0.1 \pm 0.2 \times 10^{-3}$), when taking the detected trace contents of other REE into account (NAA results given in brackets from Donovan et al. 2002).

Although ultra-violet laser wavelengths (such as 325 nm used in our study) seem to be preferred in analyses of REE emission spectra, the advantages of a selected use of excitation wavelengths were already recognised in many other studies, especially for Pr and Sm (blue, e.g. 442 nm) and for Nd and Ho (green, e.g. 532 nm) (e.g. Reisfeld et al., 1996; Gaft et al., 2005; Friis et al., 2010; Lorenz et al., 2019). Our results agree to those investigations and further highlight the specific drawbacks of non-suitable excitation conditions. For example, we observed in several of our REE-PO$_4$ samples

that non-selective or high-energy excitation (in our case especially the 325 nm laser) seem to cause a broad-band luminescence with predominant absorption features that mask potential diagnostic REE emissions at wavelengths <800 nm (see chapter 3.2). This outlines the importance of appropriate laser wavelength selection, which in turn influences the spectral detection range. Fig. 9 shows how 442 nm and 532 nm lasers can unravel REE emission lines (cf. HoPO$_4$), but also illustrates how improved REE-signal to noise ratios may be at the cost of reduced detection ranges, which may

cut-off those emission lines (cf. ErPO$_4$ in Fig. 9) at shorter wavelengths than the laser and long-pass filter wavelength, respectively.

To give an overview, we summarise our results with benefits and drawbacks for REE identification dependent on the selected laser wavelength used in this study in Figure 10. Our results emphasise the potential of selective REE excitation for technical implementation or REE differentiation and the need for careful interpretation of selectively excited LiF spectra.

Such a selective approach may deliver the needed tools in analysis of material with unknown, mixed REE contents such as in applications to natural rock.

### 4.2 Application to natural rock samples

Applications of the LiF spectral library for automated REE identifications have to deal with effects from overlaps, masking or energy transfer when several REE are present in the studied material. The presence of multiple REE may influence

relative emission intensities and also the highly variable crystal field in natural samples may influence exact emission line positions. Mixed spectra have to be expected especially from LiF of natural rocks because of the REE's similar geochemical properties and corresponding collective occurrence. Several REE seem to compete particularly due to a concentration of emission bands in the spectral range between 500 nm and 700 nm. As a result, the lines of certain REE are more likely hidden by the stronger luminescence of other REE. Therefore, an automated REE identification benefits

from several emission bands available for peak matching and needs to account for suitable excitation conditions and effects from co-occurrence.

Sm$^{3+}$, for example, shows prominent peaks at 600 nm (major) and 650 nm to be most relevant for identification, while the peak around 560 nm overlaps with the more prominent Tb$^{3+}$ emission around 545 nm (Guan et al., 2016). Similarly, the same emission lines (600 nm and 650 nm) persist, when Dy$^{3+}$ contributes to the measured spectrum and masks the

lower 560 nm emission line (Gaft et al., 2005). Pr$^{3+}$ is difficult to detect because its emissions are hidden by the lines





**Figure 8.** Graphical representation of employed excitation laser wavelengths compared to reflectance spectra from own measurements of the same REE PO$_4$ reference samples using a Spectral Evolution PSR 3500 and two published data sets of reflectance spectra (for reference details see figure legend, y-axis given in arbitrary units for illustration purposes).





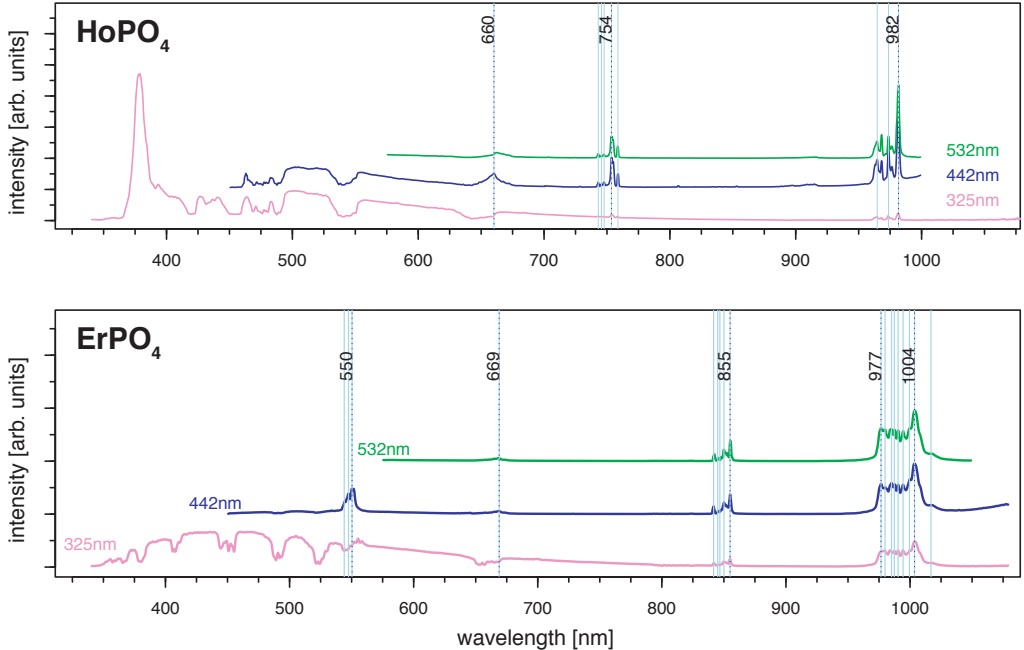

**Figure 9.** Evaluation of suitable laser wavelength for efficient REE excitation without unspecific broad-band luminescence - example HoPO$_4$ and ErPO$_4$ (dotted black lines: major emissions within emission cluster, light blue lines: emission sub-levels)

of Sm$^{3+}$ (600 - 650 nm), Dy$^{3+}$ (470 - 490 nm) and Nd$^{3+}$ (870 - 900 nm) (Gaft et al., 2005). Blue lasers being suitable for both, Pr and Sm, complicate their differentiation, while for Dy$^{3+}$ and Nd$^{3+}$ selective excitation with UV (e.g., 325 nm) or green (e.g., 532 nm), respectively, provides an option. Tm$^{3+}$ was not recorded with our system but is reported to be typically masked by Tb$^{3+}$ (Gaft et al., 2005). None of the employed laser wavelengths were successful in revealing

any diagnostic emission lines of Tm$^{3+}$, while for both, the embedded and the single grain specimen, the significant broad-band luminescence in the spectral range <700 nm dominated the spectra irrespective of 325 nm, 442 nm or 532 nm excitation. Therefore, our library cannot provide a Tm-spectrum for further applications to identify Tm in mixed spectra of natural samples. Nevertheless, we detected in all resulting spectra consistent absorption features at 355 - 365 nm, a doublet at 465 - 495 nm (most significant) and another smaller doublet 660 - 710 nm (see above Fig. 5, bottom) that agree

to diagnostic absorption features of reference samples from the USGS library (Kokaly et al., 2017). Other difficulties can arise from energy transfer effects between REE, when several REE are present as it needs to be expected in natural materials such as rocks, or from cross-relaxation phenomena for high rare-earth concentrations. Extensive research has been done, of which one simplified example are suppressed Ho$^{3+}$ emissions in the presence of Dy$^{3+}$, which can result in sensitisation of the Dy$^{3+}$ characteristic emissions at 480 nm and 580 nm (Friis et al., 2010; Lenz, 2015)





| | Laser wavelength | | |
| --- | --- | --- | --- |
| | **325nm** | **442nm** | **532nm** |
| **CePO$_4$** | (−/+) Broad-band emission distinct emission in LuPO$_4$ and YPO$_4$ | (−) Emission outside observed spectral range | (−) Emission outside observed spectral range |
| **PrPO$_4$** | (A/−) Absorptions <750nm no distinct emission | (+) Distinct emission lines | (−) Inefficient excitation |
| **NdPO$_4$** | (A/+) Absorptions <750nm distinct emission lines >750nm | (−/+) Inefficient excitation distinct emission ~900nm | (+) Distinct emission lines |
| **SmPO$_4$** | (A/+) Absorptions <750nm weak emission lines ~600nm | (+) Distinct emission lines | (+/−) Distinct emission lines few emission lines below observed spectral range |
| **EuPO$_4$** | (+) Distinct emission lines | (+) Distinct emission lines | (+) Distinct emission lines |
| **GdPO$_4$** | (−) Emission below observed spectral range | (−) Emission below observed spectral range | (−) Emission below observed spectral range |
| **TbPO$_4$** | (+) Distinct emission lines | (+) Distinct emission lines | (+/−) Distinct emission lines few emission lines below observed spectral range |
| **DyPO$_4$** | (+) Distinct emission lines | (+) Distinct emission lines | (+/−) Few distinct emission lines several emission lines below observed spectral range |
| **HoPO$_4$** | (A/+) Absorptions <700nm weak emission lines >700nm | (A/+) Absorptions <600nm distinct emission lines >600nm | (+) Distinct emission lines |
| **ErPO$_4$** | (A/+) Absorptions <750nm distinct emission >750nm | (+) Distinct emission lines | (+/−) Distinct emission lines few emission lines below observed spectral range |
| **TmPO$_4$** | (A/−) Absorptions <750nm no distinct emission | (A/−) Absorptions <750nm no distinct emission | (A/−) Absorptions <750nm no distinct emission |
| **YbPO$_4$** | (+) Distinct emission lines | No data | (+) Distinct emission lines |

(+) Successful REE emission line excitation

(−) Measurement conditions not appropriate for REE emission line excitation and/or inappropriate detection range

(A) Significant REE absorption features in broad-band emission

**Figure 10.** Summary of the evaluation of suitable laser wavelengths for an efficient excitation of individual REE (+: good conditions, -: bad conditions, A: absorption features recorded)

An application example is given in Figure 11 for a xenotime sample from Novo Horizonte. The LiF reference spectra (black lines) allow for precise REE assignment to major emission peaks in the detected mixed spectra that were obtained by 325 nm (violet line), 442 nm (blue line) and 532 nm (green line) laser excitation. Especially $Dy^{3+}$, $Sm^{3+}$ and $Nd^{3+}$ can be matched to the most prominent emission bands. The identified REE $Dy^{3+}$, $Sm^{3+}$ and $Nd^{3+}$ (cf. Fig. 11) were validated by microprobe analysis (EMPA) with 7.73 wt.%, 0.70 wt.% and 0.08 wt.%, respectively (Turner, 2015; Lorenz



385 et al., 2019). The visibility of those prominent emissions exemplifies their dependence on suitable excitation conditions (laser wavelength) and accordingly, the possibility to selectively enhance or suppress the luminescence of individual REE.

However, it is the pattern of emission bands that provides further evidence for assigning the remaining peaks to specific REE. For example, $Er^{3+}$ (EMPA: 5.26 wt.%) has an unequivocal peak above 850 nm when excited with a 442 nm laser, which may then be used as indication for its contribution and better identification in remaining emission clusters. The

390 $Er^{3+}$ luminescence apparently contributes also to the emission cluster around 1000 nm, overlapping with a potential $Yb^{3+}$ emission band (EMPA: 2.51 wt.%), as well as to several dense emission lines around 550 nm, which overlap with the $Tb^{3+}$ emission (EMPA: 1.02 wt.%) recorded during 325 nm laser excitation. The 442 nm excitation pattern further demonstrates the above mentioned competition between REE emissions (e.g. enhancement or suppression) exemplified by bright $Sm^{3+}$ emissions compared to weaker $Eu^{3+}$ emissions (EMPA: 0.12 wt.%) at partly overlapping positions. Here, the co-existence

395 of both REE ($Sm^{3+}$ and $Eu^{3+}$) seems to affect the observed emission line pattern and relative intensities shown by, for example, an intense emission peak at around 600 nm (with an orange question mark between the diagnostic $Sm^{3+}$ and $Eu^{3+}$ emission lines in Figure 11). The $Eu^{3+}$ emission lines may also be identified in the 325 nm excited spectrum (cf. results section, where all three laser wavelength showed efficient for $Eu^{3+}$ excitation), but are in the case of this sample much less intense compared to the 442 nm excited spectrum, which may be attributed to complex interactions, e.g. with

400 REE that are more efficiently excited with a 442 nm laser. Another $Eu^{3+}$ sub-level at 685 nm is, in contrast, not apparent next to the intense $Sm^{3+}$ emission cluster around 700 nm. A change in relative emission intensities is further evident in the 532 nm excited xenotime spectrum at 872 nm. The only candidate responsible for this significant peak could be $Nd^{3+}$ that has a broad emission cluster in the same range but with opposite intensity ratios. The excitation at 532 nm suggests also efficient $Ho^{3+}$ stimulation (EMPA: 1.67 wt.%), which is a potential candidate for explaining the 750 nm emission

405 overlapping with a $Dy^{3+}$ position, as well as for the 1000 nm cluster, where it is possibly mixed with $Yb^{3+}$ and $Er^{3+}$. The spectroscopic analysis of the presented phosphate rock sample from Novo Horizonte shows that the new LiF spectral library for REE phosphates enables to identify all EMPA-detected REE, despite Tm, down to a concentration of 0.08 wt%.

The observations confirm the valuable nature of reference spectra for natural sample evaluation and REE assignment. But the application example highlights at the same time that care needs to be taken with respect to complex effects

410 between REE and crystal lattice and between the, in natural minerals typically jointly occurring REE. Resulting mechanisms of energy transfer might influence relative peak intensities and complex effects from the crystal field may cause shifts of exact peak positions. Differences in relative peak intensities between assigned REE also underline the limits for quantitative conclusions. Compared to independent EMPA analysis, the $Er^{3+}$ related emissions remain weak relative to e.g. $Sm^{3+}$ and $Nd^{3+}$ during blue und green laser excitation despite its higher concentration of 5.26 wt.% versus 0.70 wt.%

415 and 0.08 wt.%, respectively. Dedicated studies are required to draw conclusions on if and how a relative quantification may be possible under which circumstances, e.g. in same host rocks from the same parent formation system.


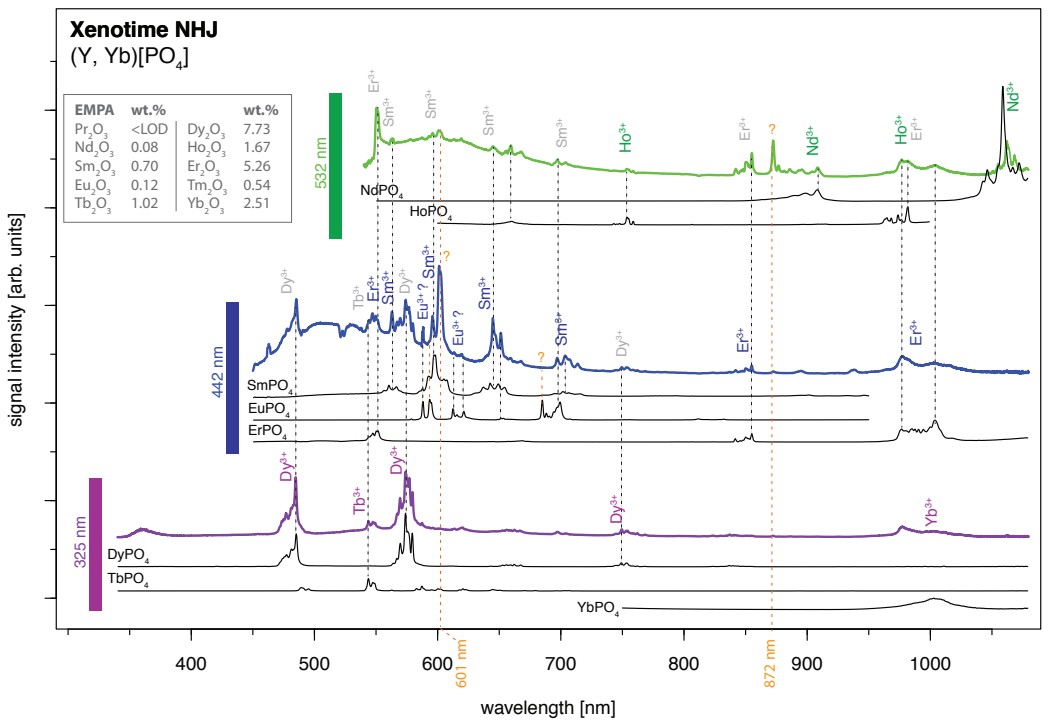

**Figure 11.** Application of the presented LiF spectral library for REE identification in a natural mineral, xenotime from Novo Horizonte, sample NHJ (see Turner (2015) and Lorenz et al. (2019))

## 5    Conclusions

We present LiF spectra for the Smithsonian REE phosphate standards in the visible to near-infrared spectral range (350 - 1080 nm). Excitation at all three commonly used laser wavelengths (325 nm, 442 nm, 532 nm) yielded spectra with distinct

REE-related emission lines for $EuPO_4$, $TbPO_4$, $DyPO_4$ and $YbPO_4$. At 325 nm excitation, most other REE-spectra were dominated by broadband defect emissions with prominent absorption features that match published reflectance data and mask expected REE emission lines. Here, lower-energy laser excitation at 442 nm showed successful especially for suppressing the non-REE-related broadband luminescence. Resulting REE reference spectra include those from $PrPO_4$, $SmPO_4$ and $ErPO_4$. For $NdPO_4$ and $HoPO_4$ most efficient excitation was achieved with 532 nm. The diagnostic emission

lines of GdPO4 lie outside the detection range and none of the three laser wavelengths was appropriate for $TmPO_4$ excitation.

Our results demonstrate the suitability of LiF for REE detection and especially the possibility of selective element excitation. Our reference spectra provide the full spectral information at high resolution (0.13 nm) as a basis for an improved evaluation of REE-bearing natural rocks. The spectral library allows for analysis of emission line positions, emission

line intensity ratios and splitting into emission line sub-levels and for transparent comparison with published and future LiF measurement data. The permanent storage and open access of the data via the online repository zenodo.org and registered DOI: 10.5281/zenodo.4054606 (**?**) provide the needed basis for sharing and re-using the data set. Thus, the available spectral data support the use of LiF for REE analysis in natural samples and its application in raw material explo-ration. Applications in natural rock material rely on the suppression of potentially strong matrix emissions. In this respect,

selective laser wavelength instead of high power UV stimulation and time resolved measurements promise considerable improvements for reliable REE identification. The integration of reflectance hyperspectral imaging provides great potential for cross-validation of REE identification and for insights into mineralisation processes.

## 6  Data availability

The data comprising the LiF spectral library for REE orthophosphates are available via the online repository zenodo.org,

where the data is permanently stored and registered with the DOI: 10.5281/zenodo.4054606 (Fuchs et al., 2020). The LiF library gives open access to the spectral data under the Creative Commons Attribution 4.0 International license. You can cite all versions of the data set provided via the LiF spectral library for REE orthophosphates at zenodo.org by using the given DOI to refer to Fuchs et al. (2020) and citing this article.

*Author contributions.* Margret C. Fuchs and Jan Beyer designed and performed all experiments for data acquisition in the optical

characterisation laboratory of the TU Bergakademie Freiberg. Scientific results and corresponding interpretations were extensively discussed by Margret C. Fuchs with all co-authors regarding physical background (Jan Beyer, Suchinder Sharma, Johannes Heitmann) and their mineralogical and geological implications (Sandra Lorenz, Axel D. Renno and Richard Gloaguen). Margret C. Fuchs prepared the manuscript with contributions from all co-authors.

*Competing interests.* No competing interests are present.

*Acknowledgements.* We thank the Smithsonian Institution Department of Mineral Sciences and Tim Rose for providing and send-ing the REE phosphate samples NMNH 168484 - 168499 used in this study (see also https://naturalhistory.si.edu/research/mineral-sciences/collections-overview/reference-materials/smithsonian-microbeam-standards). The scientific work was funded by EIT RawMa-terials as part of the upscaling project inSPECtor (grant no. 16304). Data acquisition was done in the optical characterisation laboratory of the Institute of Applied Physics at the TU Bergakademie Freiberg.



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
