# Peer review of "A spectral library for laser-induced fluorescence analysis as a tool for rare earth element identification"

_Earth System Science Data, 2020_

## Referee Comment (RC1) · Uwe Altenberger (Referee) · 8 Jan 2021

The submitted work is a contribution to a highly interesting and important topic. Rare earth elements (REE), especially the heavy ones are still urgently needed; a dependency of the raw materials on a few countries has accelerated global exploration. The "fast" analysis of these elements, e.g. in drill cores, rock walls, alluvial sediments etc. is therefore of particular importance. The present manuscript framing the already stored dataset is focused on the identification of rare earth element by laser-induced fluorescence analysis (LiF). The working approach is the systematic analysis of pure REE phosphates, which in turn represent standards for EMP analyzes. The aim is to

qualitatively record natural minerals from the results obtained. Despite promising, published work, there is a lack of systematic LiF studies of REE phosphates in the area of raw material acquisition. The presented work is supposed to fill this lack and does it. The work is structured logically and consistently, the methodology and results are well explained and documented, even for readers who are not specialized. There is a clear distinction to more recent work (e.g. Seidel et al. 2019), in which authors of the present study were involved. The new and positive thing about the present work is the systematic approach and the use of three lasers (325, 442, 532nm). Seidel et al. used only two lasers. In addition, the recording over a very large spectral range (340-1080nm) makes the work important, too. The selection of the samples is of particular importance, they are international standards (Smithsonian EMPA standards). In addition to embedded samples, single grains are also analyzed, which is important as embedding agents create overlapping. The description of the laser used and the application to the individual REE phosphates is detailed, clear and easy to understand for users. The investigations are performed meticulous. To my knowledge, published and cited LiF works and data collections for naturally occurring samples are not so systematic, element-selective, of the same high resolution and methodically differentiated. The work ends with the application to a naturally occurring HREE phosphate (Xenotime). A supplement to other REE minerals, such as REE carbonates, which also occur frequently, would be desirable. This is a suggestion for future work, not a criticism of the present work. The significance of the submitted manuscript is extremely high. It is a very helpful tool and unique in its completeness. It represents the basis for various options, especially in the raw material sector. For example, a relatively quick scanning of drill cores, handpieces or open pits can test the material for increased REE contents. The utility is then primarily an applied one. However, the data set can also serve as a basis for further research, e.g. the recording of carbonaceous REE minerals

The data and images described in the manuscript correlate perfectly with the data set that has already been saved - this means that they can be checked or used for other applications. The summarizing figure (Fig. 10), will help the user choosing the best
laser wavelength for the subject. A specific quantitative analysis is not given, yet. But this is beyond the scope of the project

In my opinion, this is a very professional study. The choice and combination of different lasers and samples (artificial mounds and single grains as well as natural samples) is excellent and really brings new results. Again, the text is extremely well written and very understandable and comprehensible for LiF non-professionals. Text and database correlate very well. Except for some small orthographic errors, I don't see any point of criticism.

I rate the presented work with 1

To be checked/corrected Abstract: line 3 I would add . . .and minerals to . . ..such as rocks. . .. p. 2 line 64: ..are instead of is?

p.4: line 42/43: . . . multiplied by 0. I am not sure if 0 is ok

---

## Referee Comment (RC2) · Anonymous Referee #2 · 18 May 2021

My opinion in manuscript essd-2020-296

Fuch et al., "
*A spectral library for laser-induced fluorescence analysis as a tool for rare earth element identification*"

General comments:

The search for REE elements using non-destructive methods is an important issue, also in the economic aspect. A quick and easy method of assessing the mineral content of REE in the drill core is desirable. The authors propose such a method. Therefore, undertaking such a research task is positive. I understand and approve need to obtain a quick and relatively easy answer to the question of whether there are REE in the natural sample, i.e., in a mineral or in a rock. I understand and approve the idea of creating a digitized base of the luminescence spectra of the ions of these elements. All luminescence researchers have some form of base. It also includes references, which the authors call "paper" ones. Among them are works – books (for example Blasse& Brabmair, Springer) or websites, easy to check.

I am glad to read an article promoting the possibilities of the luminescence method. You have to be very careful when interpreting the luminescence spectra. The authors of this work know this. However, I believe that they should make it clearer in the reviewed article that neither the base they created, nor the tables or charts of this work will make beginner researchers take measurements and interpret their results correctly.

I cannot define the target audience of this work. If the group of readers is to be wide - judging from the profile of the journal - then you need to be not only substantive, but also in addition to the advantages of this method, very clearly indicate its limitations. The recommendations in Figure 9 will not suffice. If this work is to assist a wide range of drill developers, those who have not had much experience with luminescence measurements, readers should also be alerted to how samples are prepared for measurement. I don't know if they will be separate grains or cuts. In the latter case - it would be a reflective measurement, i.e., different than for base materials. I do not know if such reflective luminescence spectra were made - you can only see absorption ones.

I also checked the functionality of the database. The compatibility of the database files with own files from measurements made on two different devices was checked. As might be expected, the main, sometimes big problem was to choose the right scale of the Y axis to check whether or not the emission line is overlapping. When superimposing several spectra from the base and several measurement spectra, the problem was very time-absorbing. It was hard for me to notice any clear advantage over the classical method. But - maybe it's a matter of the employee's age and habits.

Positive detailed comments

1. the authors know the commonly measured effect that the emission bands of 4f ions are often split into components and that the emission may come from several closely spaced energy levels (Pr3 +);

2. the measurements that have been taken are summarized in the figures 3-7 and in Tables 1-4, in particular in Figure 9, is a summary for workers who are starting their research on lanthanide luminescence;

3. the spectra (files) are available through reference *Fuchs et al., 2020* – as .txt files, privately, I do not like such files, but they can be used. Used and compared with own data.

4. the reviewed study indicates to the future researcher many cases that require additional research - e.g. the presence of Sm(3+) beside to Eu(3+), and Pr(3+) beside to Sm(3+). It has rightly been noted that the 611nm -615nm line from Eu(3+) may be missed if Sm3 + is also present in the sample. However, the excitation spectra of the two ions differ, so it should be recommended to perform them;

5. the proposed set of measured spectra may be applicable and helpful, but be aware of its limitations.

Some critical remarks:

1. In my opinion, the material prepared as in the reviewed article cannot be called "the base".

2. The Authors' results allow only a fairly approximate verification of the presence of these ions in the tested sample.

3. Only 3+ ions were tested, but what about the ubiquitous Eu2+? between Ce(3+) and Eu(2+) there is often an energy transfer and the band from Eu(2+), very often in 415-460 nm range, is more intense than from Ce?

4. The photoluminescence spectrum of the tested sample will show the presence of the center, in this case the $4f^n$ ion, when it is excited to glow with the appropriate energy. In the case of this article - a laser line of the appropriate length. This is a basic and known fact.
   What is shown in Figures 3-8 and in Tables 2, 3, 4, is commonly known from the "paper" literature, the references that the authors of this work wish to escape from.

5. The measurements were made for standards of only one lanthanide phosphate.  And yet, the presence of several lanthanide ion together in minerals, as is usual, may cause problems in their unambiguous identification. In my opinion, the automation of digitized measurements proposed by the authors may lead to numerous errors. It must be remembered that the 4f ions that appear next to each other can transmit the emitted energy (energy transfer effect), in a sense obscure each other. In this situation, it is of great help to perform excitation measurements for individual emission lines. This problem was mentioned in the reviewed work quite briefly, and it is very complex. Even the measurements shown in Fig. 8 show how much excitement measurements are needed. For this reason, the spectra in the earlier Figures 3-7 and Tables 2-4 cannot be considered sufficient for the correct / final identification of the 4f ions. Measurements made for excitation with 3 laser lines can actually be very helpful in checking the presence of some lanthanide ions in the tested sample, but in my opinion, they will not replace the full / fuller spectroscopic characteristics of the test object. Although the wavelength of emission lines of RE ions, apart from Eu(2+) and Ce(3+), weakly depend on the strength of the crystal field, their exact value sometimes changes depending on the matrix. An example is Yb (3+) - see literature and figure at the bottom of the review. One can therefore overlook one ion, especially when it is present in the sample in small amounts alongside another one with more intense luminescence. Moreover, the intensity of some Eu(3+) transitions depend on the site symmetry and could be very weak.

6. The luminescence efficiency of individual ions in the same matrix is different. Comparing the spectra on an arbitrary scale may introduce the (younger) researcher to erroneous conclusions. For example, for the excitation of 323 nm and 1: 1 proportion of Dy and Eu (0.1%) in the phosphate glass, the emission bands from Dy are much stronger than the Eu (3+) bands; actually, a very weak band at 611 nm is visible. Using only the 325 nm laser excitation, it cannot be concluded that Eu (3+) is present in the tested sample.

7. There is no indication of the detection limit of the ion; this can be checked in previously published (paper) articles, for example for phosphate glasses subsidized with 4f ions.

8. The results and "base" can certainly be treated as an auxiliary statement.

9. Of course, authors can only measure the available laser lines. Therefore, for precise identification tests, I will recommend other measurements, as in the reference Czaja et al., (2013) Journal of Mineralogical and Petrological Sciences, 108, 47-54, https://doi.org/10.2456/jmps.111229.

Some minor / detailed comments:

1. Line 170: I do not know the mineral for which Ce(3+) emission band has been measured at 540 nm . I purpose to delete this value. The Ce (3+) is of great economic importance. Perhaps it should be written a little more clearly that the position of the emission band of this may change depending on the material / mineral in which it occurs.

2. Line 250-255: Sm(2+) luminescence is measured ONLY at low temperature (see, for example Gaft et al., 2005 page 142)

3. The $4f^n$ ions often substitute for Ca (2+). However, Mn2 + is quite common in Ca minerals, so in the measured spectra gives an intense emission band, from about 600-700 nm. It often obscures emissions from Sm(3+), Eu(3+) and Pr(3+). So - how to get out of such a situation? In many minerals containing $4f^n$ ions, luminescence can be caused by other centers, not only Mn(2+), also anions of $WO_4$ $MoO_4$ $CrO_4$, $TiO_6$ complexes, defective oxygen and others. The emission from these ions may obscure the emission from the lanthanides.

4. The excitation band at 442 nm is favored primarily for Pr(3+), and to some extent also for Sm (3+). In order to distinguish between these ions, it is more advantageous to use different excitation wavelengths, also, by the way, not to lose Eu(3+). However, I am not sure if this line efficiently excites Er(3+) (Fig. 6 and Table 3). According to my (and other reference) data, the emission intensity of Er(3+) is more than 3 times weaker for this excitation, compared to the excitation of 377 nm.

5. Please explain why in figure 9 it is indicated that the distinct emission bands from Eu (3+) were measured at each of the 3 excitation lines? earlier figures/tables do not confirm this.

6. Fig. 11 - I believe that the bands marked in yellow and with a question mark are surely the bands from Sm3 +; in my opinion, the question mark can be removed.

---

## Author Comment (AC1) · 15 Jun 2021

Response to reviewer's comments
Thank you very much for this positive feedback and your recommendations! We are currently working on several additional aspects concerning efficient technical solutions as well as testing applications to further mineral hosts and their respective geological settings. We also test further integrations of optical technologies to take full advantage of their benefits and use complementary information (e.g. already was published by Lorenz et al., cited in this manuscript). Integrative solutions may also address the quantitative dimension. However, this will demand for smart solutions to keep industrial applications realistic and add value to existing technologies by providing a noninvasive, ecologic and economically efficient alternative for exploration operations.

To be checked/corrected

Abstract: line 3 I would add ...and minerals to ....such as rocks. . ..

ok, we added "Applications to natural materials such as minerals and rocks . . ."

p. 2 line 64: ..are instead of is?

Unfortunately, we cannot find a matching phrase for this comment. We changed the sentence originally on p. 3 l. 61-64 and separated into two to hopefully make the meaning more clear: "Nevertheless, the availability of new, sophisticated, automated data processing routines emphasises already today the need for digital reference data of complete spectra, comparable to those for HSI (e.g. Kokaly et al., 2017) or Raman spectroscopy (e.g. Lafuente et al., 2015). Such reference data are crucial to further develop and automate the LiF-based REE detection and analytical capacities for analyses of REE abundances and their spectral representation in natural rocks."

p.4: line 42/43: ... multiplied by 0. I am not sure if 0 is ok

We assume that the comment refers to p. 6 line 125, where the multiplication by zero applies to the first data point of the merge segment only. For further data points, the weighting factor linearly increases to one across the merge segment to gradually transition from the first to the second spectrum. This allows for a smooth merge without

artificial steps in the spectra which could corrupt analytical detection methods. So mathematically, the multiplication with zero is ok, but we acknowledge that the formulation was somewhat complex. Thus, we suggest the following new formulation: "Within the merge segment, the final spectrum transitions smoothly from the first (measured with 334 nm long-pass filter, covering the short wavelength range) to the second partial spectrum (measured with 550 nm long-pass filter covering the long wavelength range). Mathematically this means, that the difference of the partial spectra multiplied by a linearly varying weighting factor from zero to one across the merge segment is added to the first spectrum. Finally, the partial spectra were ..."

---

## Author Comment (AC2) · 15 Jun 2021

My opinion in manuscript essd-2020-296 Fuch et al., " A spectral library for laser-induced fluorescence analysis as a tool for rare earth element identification"

General comments: The search for REE elements using non-destructive methods is an important issue, also in the economic aspect. A quick and easy method of assessing the mineral content of REE in the drill core is desirable. The authors propose such a method. Therefore, undertaking such a research task is positive. I understand and

approve need to obtain a quick and relatively easy answer to the question of whether there are REE in the natural sample, i.e., in a mineral or in a rock. I understand and approve the idea of creating a digitized base of the luminescence spectra of the ions of these elements. All luminescence researchers have some form of base. It also includes references, which the authors call "paper" ones. Among them are works – books (for example Blasse& Brabmair, Springer) or websites, easy to check.

Review comment:

I am glad to read an article promoting the possibilities of the luminescence method. You have to be very careful when interpreting the luminescence spectra. The authors of this work know this. However, I believe that they should make it clearer in the reviewed article that neither the base they created, nor the tables or charts of this work will make beginner researchers take measurements and interpret their results correctly.

Reply:

We agree with the reviewers comment that careful interpretation of luminescence spectra is needed for meaningful results. The goal of this study is to provide objective reference data as it is already state of the art in several other spectroscopic disciplines such as hyperspectral imaging and Raman spectroscopy, of which especially the earlier experienced a boost in analytical capacities from available spectral libraries, global or site specific. Others such as LIBS, X-ray diffraction analysis or X-ray fluorecence analysis come often with internal libraries in measurement devices. In the case of the LiF method, further development requires access to reference data, for which the presented library can only provide a start, far from complete, but open to be complemented with other host data or data with site specific variations.

We added to the text (conclusions): "Multiple complex interactions between naturally co-existing REE and with complex variable host configurations in natural rocks still require careful interpretation of LiF spectra and a good understanding of luminescence principles."

Review comment:

I cannot define the target audience of this work. If the group of readers is to be wide - judging from the profile of the journal - then you need to be not only substantive, but also in addition to the advantages of this method, very clearly indicate its limitations. The recommendations in Figure 9 will not suffice. If this work is to assist a wide range of drill developers, those who have not had much experience with luminescence measurements, readers should also be alerted to how samples are prepared for measurement. I don't know if they will be separate grains or cuts. In the latter case - it would be a reflective measurement, i.e., different than for base materials. I do not know if such reflective luminescence spectra were made - you can only see absorption ones.

Reply:

The target audience is primarily in the field of exploration, and in particular the developers of advanced data processing routines for spectral analysis, but also may be seen in further research fields working on the REE luminescence in natural rocks or artificial compounds.

We added to the text in the abstract: "Primarily addressing the raw material exploration sector, it aids particularly the development of advanced data processing routines for LiF analysis, but also can support further research on the REE luminescence in natural rocks or artificial compounds."

... and in the last paragraph of the introduction: "Our study aims to provide a useful database for mineral characterisation in LiF-based REE exploration, where particular target groups are developers of advanced data processing routines and of technical implementations, but also various scientists working on further aspects of REE luminescence. "

With our focus on data analysis and processing, material preparation lies beyond the

scope of our study. Concerning the mode of measurement, we like to point out that typically luminescence measurements are conducted in backscattering mode, where the emitted luminescence is collected in the same or at least almost same direction as from where the excitation light came. In this configuration all presented measurements were taken (see Fig. 2 in the manuscript). We added a sentence to chapter 2.2: "The configuration represents a typical luminescence measurement setup in backscattering mode, where the light paths of excitation to the sample and emitted luminescence from the sample are parallel."

We further change the conclusions to highlight the limitations regarding 1) focus on trivalent REE and need for complementary data, 2) possible matrix luminescence and option of selective excitation, where the three laser wavelengths provide an option, but scientific in-depth analysis requires more flexible equipment, and 3) need for expert knowledge to interpret the mixed spectra resulting from complex energetic interactions in natural rocks.

Review comment:

I also checked the functionality of the database. The compatibility of the database files with own files from measurements made on two different devices was checked. As might be expected, the main, sometimes big problem was to choose the right scale of the Y axis to check whether or not the emission line is overlapping. When superimposing several spectra from the base and several measurement spectra, the problem was very time-absorbing. It was hard for me to notice any clear advantage over the classical method. But - maybe it's a matter of the employee's age and habits.

Reply:

Thank you for this comment and directly trying the usability of our library data! We recommend here using programming code based analysis tools that would reduce the efforts to a few (1-2) lines of code to scale the reference spectrum to the observed data. As stated in above review responses, our target audience are in particular developers

of data processing tools, which then would increase efficiency in analysis of large data sets as required e.g. in the case of drill-core scans.

Positive detailed comments

Review comment:

1. the authors know the commonly measured effect that the emission bands of 4f ions are often split into components and that the emission may come from several closely spaced energy levels (Pr3 +);

Reply:

Thank you. This agrees to our intension to make such data available for further use and comparison.

Review comment:

2. the measurements that have been taken are summarized in the figures 3-7 and in Tables 1-4, in particular in Figure 9, is a summary for workers who are starting their research on lanthanide luminescence;

Reply:

Thank you. Besides describing our data set in detail, the summary of lanthanide luminescence in a more general sense will hopefully support also new users.

Review comment:

3. the spectra (files) are available through reference Fuchs et al., 2020 – as .txt files, privately, I do not like such files, but they can be used. Used and compared with own data.

Reply:

Thank you. We chose the .txt file format because it can easily be loaded into many programming environments.

Review comment:

4. the reviewed study indicates to the future researcher many cases that require additional research - e.g. the presence of Sm(3+) beside to Eu(3+), and Pr(3+) beside to Sm(3+). It has rightly been noted that the 611nm -615nm line from Eu(3+) may be missed if Sm3 + is also present in the sample. However, the excitation spectra of the two ions differ, so it should be recommended to perform them;

Reply:

Thank you. We discuss such energy transfers between REE more in detail using the application example (see chapter 4.2).

We added the reviewers note in an additional sentence: "The intensified diagnostic Sm3+ versus relatively weak Eu3+ emission lines suggest an energy transfer. The energetic transfer to Sm reduces particularly the Eu3+-diagnostic emission lines at 613 - 622 nm to very low signal intensities. In the 325 nm excited spectrum, those Eu3+ emission lines at 613 - 622 nm may be present as weak signals as well as those around 700 nm, but the prominent Dy3+ lines preclude an unequivocal identification."

and also changed the section to emphasise further on details observed in the example case illustrated in Fig. 11.

The value of performing measurements with different excitation spectra is discussed in the 2nd paragraph of the application example chapter 4.2., e.g. with: "... Blue lasers being suitable for both, Pr and Sm, complicate their differentiation, while for Dy3+ and Nd3+ selective excitation with UV (e.g., 325 nm) or green (e.g., 532nm), respectively, provides an option. ..." and further on.

Review comment:

5. the proposed set of measured spectra may be applicable and helpful, but be aware of its limitations.

Reply:

Thank you for this comment. We exemplify limitations using the application case in chapter 4.2 and point at the need for expert knowledge and in-depth analysis in research in our conclusions.

Some critical remarks:

Review comment:

1. In my opinion, the material prepared as in the reviewed article cannot be called "the base".

Reply:

We use the term in the sense of a base for comparison.

To clarify, we changed in conclusion "base" to "objective database" and "needed basis" to "needed functionality"

Review comment:

2. The Authors' results allow only a fairly approximate verification of the presence of these ions in the tested sample.

Reply:

As the reviewer stated above, the data alone cannot replace the need for expert knowledge, but the example illustrates how the REE library can provide the data base for comparison may it be for manual/visual comparison or for automated processing approaches. The goal is, to identify the potential candidates responsible for the observed emission bands.

We added in conclusions "However, usage of the library for visual comparison or implementation into innovative automated data analysis routines facilitates identification of most probable REE candidates responsible for observed luminescence also in cases

of similar emission lines, but cannot replace the need for expert knowledge."

Review comment:

3. Only 3+ ions were tested, but what about the ubiquitous Eu2+? between Ce(3+) and Eu(2+) there is often an energy transfer and the band from Eu(2+), very often in 415-460 nm range, is more intense than from Ce?

Reply:

We agree with the reviewer's comment that in applications to rocks and minerals energy transfer is an important issue to be considered! We already had included in the initial submission (first paragraph of application example in discussion, chapter 4.2):"Applications of the LiF spectral library for automated REE identifications have to deal with effects from overlaps, masking or energy transfer when several REE are present in the studied material. The presence of multiple REE may influence relative emission intensities and also the highly variable crystal field in natural samples may influence exact emission line positions."

We added to conclusions (see response to general comments above): "Multiple complex interactions between naturally co-existing REE and with complex variable host configurations in natural rocks still require careful interpretation of LiF spectra and a good understanding of luminescence principles."

We also agree with the reviewers point that several REE of different valence such as Eu2+ are relevant for application cases. Here with this contribution, we started with the well described set of trivalent REE, but as we state in the manuscript, the library will benefit largely from additional data.

We added to the conclusions: "Yet, the presented library is limited to trivalent REE hosted in orthophosphates. Complementary spectral data of other economically relevant REE configurations (e.g. divalent REE or other host minerals) are needed to further improve LiF capacities in REE exploration."

Review comment:

4. The photoluminescence spectrum of the tested sample will show the presence of the center, in this case the 4fn ion, when it is excited to glow with the appropriate energy. In the case of this article - a laser line of the appropriate length. This is a basic and known fact.

Reply:

As already stated by the reviewer, the circle of readers of the article cannot be sharply defined, although from our perspective the main target group is found in the field of raw material exploration and mineral characterisation. We are of course aware that this must lead to minor repetitions of textbook knowledge, but we consider this unavoidable in the interest of readability for a wide range of potential readers. Additionally, we believe the presentation of a new spectral library data set requires a solid description of the materials luminescence properties. Based on that, our application example serves as a show case to illustrate how the reference spectra in combination with the three different laser wavelengths can support differentiation of REE.

Review comment:

What is shown in Figures 3-8 and in Tables 2, 3, 4, is commonly known from the "paper" literature, the references that the authors of this work wish to escape from.

Reply:

We present the Fig. 3-8, because those are the measurement results associated with the published reference data and therefore, are included to document the detailed luminescence properties of the reference material. We see it as a positive fact and proof of the quality of the reference material that the data agrees to published literature. We further give reference to the published work that represents the respective literature and hence, believe that we link our work clearly and transparently to the state of the art. We would further like to note that we consider the scientific information as the

decisive criterion and not the way of its presentation. However, the electronic/digital availability of verified information allows a faster integration into automated processes and reduces the risk of transmission errors and knowledge loss.

Review comment:

5. The measurements were made for standards of only one lanthanide phosphate. And yet, the presence of several lanthanide ion together in minerals, as is usual, may cause problems in their unambiguous identification. In my opinion, the automation of digitized measurements proposed by the authors may lead to numerous errors. It must be remembered that the 4f ions that appear next to each other can transmit the emitted energy (energy transfer effect), in a sense obscure each other. In this situation, it is of great help to perform excitation measurements for individual emission lines. This problem was mentioned in the reviewed work quite briefly, and it is very complex. Even the measurements shown in Fig. 8 show how much excitement measurements are needed. For this reason, the spectra in the earlier Figures 3-7 and Tables 2-4 cannot be considered sufficient for the correct / final identification of the 4f ions. Measurements made for excitation with 3 laser lines can actually be very helpful in checking the presence of some lanthanide ions in the tested sample, but in my opinion, they will not replace the full / fuller spectroscopic characteristics of the test object. Although the wavelength of emission lines of RE ions, apart from Eu(2+) and Ce(3+), weakly depend on the strength of the crystal field, their exact value sometimes changes depending on the matrix. An example is Yb (3+) - see literature and figure at the bottom of the review. One can therefore overlook one ion, especially when it is present in the sample in small amounts alongside another one with more intense luminescence. Moreover, the intensity of some Eu(3+) transitions depend on the site symmetry and could be very weak.

Reply:

Thank you for this thoughtful comment. The focus of our paper presenting a spectral

REE library cannot cover all the effects occurring in natural rocks, but we agree that careful evaluation and no straight simplified REE identification is possible in many applications. As replied to your general comment above, we added text to the conclusions to emphasise on the complex interactions between naturally co-existing REE and the need for expert knowledge.

Review comment:

6. The luminescence efficiency of individual ions in the same matrix is different. Comparing the spectra on an arbitrary scale may introduce the (younger) researcher to erroneous conclusions. For example, for the excitation of 323 nm and 1: 1 proportion of Dy and Eu (0.1%) in the phosphate glass, the emission bands from Dy are much stronger than the Eu (3+)bands; actually, a very weak band at 611 nm is visible. Using only the 325 nm laser excitation, it cannot be concluded that Eu (3+) is present in the tested sample.

Reply:

We assume, this comment refers to Fig. 11. Yes, it is exactly what we observe, the 325 nm laser excitation results in much stronger Dy emissions and Eu(3+) is only distinguishable during 442 nm laser excitation. Arbitrary units are necessary because of the offset of the spectra from different laser excitation wavelengths and for better visualisation of the observed emission lines in individual spectra. Also the reference spectra are scaled to illustrate the potentially matching peaks.

We added a comment to the figure captions: "The measured spectra from the three different laser excitation wavelengths are offset and luminescence intensities of individual spectra are scaled for better visualisation of observed emission lines and potential reference emission peaks (arbitrary units: comparison of signal intensities between spectra is not meaningful)."

We also added to the text for discussing the application example: "In the 325 nm excited

spectrum, those Eu3+ emission lines at 613 - 622 nm may be present as weak signals as well as those around 700 nm, but the prominent Dy3+ lines preclude an unequivocal identification."

Review comment:

7. There is no indication of the detection limit of the ion; this can be checked in previously published (paper) articles, for example for phosphate glasses subsidized with 4f ions.

Reply:

Although we recognize that questions of detection limits of individual ions are of high scientific value, they did not play a decisive role in the current state of our investigations, nor do we see any direct impact at the moment on the goals we have set for our presented study.

As discussed above, the luminescence intensity of some RE ions depends strongly on the particular lanthanide content of every sample, due to e.g. energy transfer. Furthermore, each particular luminescence setup will have differing detection efficiencies and signal-to-noise-ratios, which together with different sample surface roughnesses, affects strongly such values. Thus we note, that any such claims of detection limits might be of limited value to the reader, which is why we did not investigate detection limits in our study. Nevertheless, in further investigations on case studies with REE enriched rocks this may represent a decisive factor, and we will of course take this into account as well as the literature named by the reviewer.

Review comment:

8. The results and "base" can certainly be treated as an auxiliary statement.

Reply:

We assume this corresponds to critical remark no. 1. We changed the wording. For

details, please see our corresponding response above.

Review comment:

9. Of course, authors can only measure the available laser lines. Therefore, for precise identification tests, I will recommend other measurements, as in the reference Czaja et al., (2013) Journal of Mineralogical and Petrological Sciences, 108, 47-54, https://doi.org/10.2456/jmps.111229

Reply:

We agree that most efficient REE excitation may be achieved with wavelengths other than the here employed ones, but this was not the scope of our study. We aimed at reducing excitation efforts to a reasonably limited number of wavelengths and evaluate how well REEs can be determined. This is useful in terms of future implementation in raw material exploration and development of respective technical devices.

We added to the introduction (compare reply to the 2nd general comment): "Our study aims to provide a useful database for mineral characterisation in LiF-based REE exploration, where particular target groups are developers of [. . .] and of technical implementations . . .,"

One of the goals was listed already to be "(2) investigating the suitability of three standard laser wavelengths (325 nm, 442 nm, 532 nm) with respect to excitation efficiency and selectivity, " and we added to conclusions: " The three investigated laser wavelength may provide the ground for robust technical solutions of a LiF sensor in exploration. "

We are aware of the publications from Czaja et al. and also cited two of them in the presented manuscript as well as used the CSIRO luminescence database, where Czaja et al is cited. Additionally, we added a to the conclusions: "Detailedexcitation spectra as well as time-resolved luminescence measurements represent valuable options for further insights into the causes and mechanisms behind the observed spectra (e.g. Czaja

et al., 2013; Gaft et al., 2005).." to account for the need of more specific excitation conditions on most research applications.

Some minor / detailed comments:

Review comment:

1. Line 170: I do not know the mineral for which Ce(3+) emission band has been measured at 540 nm . I purpose to delete this value. The Ce (3+) is of great economic importance. Perhaps it should be written a little more clearly that the position of the emission band of this may change depending on the material / mineral in which it occurs.

Reply:

We changed the sentence to highlight the maximum at 420 nm (540 nm was only the upper range limit of the full width half maximum estimate, we removed the value): "For CePO4, the excitation results in a broadband emission with a maximum at 420 nm (Fig.4)"

Review comment:

2. Line 250-255: Sm(2+) luminescence is measured ONLY at low temperature (see, for example Gaft et al., 2005 page 142)

Reply:

Thank you for this comment! This clarifies also, why the 734 nm emission line is not observed.

We changed the sentence: "Influences from Sm2+ centres are considered to be unlikely because of measurements at room temperature and absence of the 734 nm emission line (Gaft et al., 2005)"

Review comment:

3. The 4fn ions often substitute for Ca (2+). However, Mn2 + is quite common in Ca minerals, so in the measured spectra gives an intense emission band, from about 600-700 nm. It often obscures emissions from Sm(3+), Eu(3+) and Pr(3+). So - how to get out of such a situation? In many minerals containing 4fn ions, luminescence can be caused by other centers, not only Mn(2+), also anions of WO4 MoO4 CrO4, TiO6 complexes, defective oxygen and others. The emission from these ions may obscure the emission from the lanthanides.

Reply:

Thank you for this valuable comment! - We assume this refers mainly to application cases, where the mentioned anions may play a role in observed spectra, while they were not reported for the used reference material (see Donovan et al. 2002).

We added the information to the application example, added the comment that respective luminescence may cause more broadened peaks, time-resolved luminescence measurements may overcome the problem of obscured REE peaks and highlight the need for corresponding reference spectra: "Additionally, we want to emphasise on the fact that anion sites in minerals, typically substituted by REEs, also can be occupied by other luminescent anions (Gaft:2005). For example, Mn2+ substitutes for Ca2+ or Mg2+, Cr3+ for Al3+ and others. Also complexes with oxygen such as WO4, MoO4, CrO4 and TiO6 may contribute to the observed spectra and then may obscure the REE emission lines at overlapping wavelengths positions. In those cases, complementary reference spectra for the corresponding anions and complexes would help a correct identification, while more detailed luminescence measurements such as time-resolved detection are needed to unravel REE emissions behind the often broader peaks such as from Mn2+."

Review comment:

4. The excitation band at 442 nm is favored primarily for Pr(3+), and to some extent also for Sm (3+). In order to distinguish between these ions, it is more advantageous

to use different excitation wavelengths, also, by the way, not to lose Eu(3+). However, I am not sure if this line efficiently excites Er(3+) (Fig. 6 and Table 3). According to my (and other reference) data, the emission intensity of Er(3+) is more than 3 times weaker for this excitation, compared to the excitation of 377 nm.

Reply:

Our data showed that UV excitation at 325 nm caused a broad band emission masking the emission band around 550 nm that is also described in the literature for e.g. apatite (e.g. Gaft et al. 2005, page 163/164 ) depicted in Fig. 9. The excitation at 442 nm showed successful to unravel the previously hidden emission band and compared to absorption characteristics shown in Fig. 8, the laser line at 442 nm coincides with an absorption feature of $Er^{3+}$ suggesting suitable conditions. We also show in Fig. 9 that excitation at 532 nm is also successful, but cannot record the 550 nm emission band due to filter setting to separate the laser line from the recorded spectrum.

For clarity, we add to the text (discussion chapter 4.1): "For example in the case of Er3+, efficient laser excitation lines are commonly reported to be in the UV or green spectral range (e.g. Gaft et al., 2005). Our results, however, show a broad-band emission when using a UV laser that obscures the emission band around 550 nm. The use of the 442 nm laser excitation enables to record this emission, which would remain at wavelengths shorter than the detection range, when using a 532 nm laser, although the latter shows as well suitable excitation conditions. Comparison the absorption characteristics depicted in Fig. 8 outlines that the 532 nm laser is close to a very efficient excitation band and hence may be preferred in many application cases, but also the 442 nm laser can provide suitable excitation conditions."

Review comment:

5. Please explain why in figure 9 it is indicated that the distinct emission bands from Eu (3+) were measured at each of the 3 excitation lines? earlier figures/tables do not confirm this.

Reply:

We assume that the comment refers to Fig. 10 (table) and to the application case with Fig. 11., while in earlier figures/tables, Eu3+ is only mentioned in the Fig. 4 and Tab.2. The initial text states that for Eu3+ all three laser wavelengths were successful, e.g. in conclusions: "Excitation at all three commonly used laser wavelengths (325 nm, 442 nm, 532 nm) yielded spectra with distinct REE-related emission lines for EuPO4, 420 TbPO4, DyPO4 and YbPO4" and at the beginning of chapter 3.3 on blue laser excitation). This agrees to the summary given in Fig. 10.

We added for clarity a sentence to chapter 3.4 on green laser excitation: "When using 532 nm laser excitation, the diagnostic emission lines already described above for 325 nm excitation (see chapter 3.1) were recorded for the samples NdPO4, EuPO4, TbPO4, DyPO4. "

For the application case, we already discuss the weak emission lines of Eu3+ in the initial submission. Nevertheless, we changed the section to incorporate also comments from above and emphasise on why the three lasers seem not to be efficient for Eu excitation in the application example in the discussion chapter 4.2.: "The intensified diagnostic Sm3+ versus relatively weak Eu3+ emission lines suggest an energy transfer. The energetic transfer to Sm3+ reduces particularly the Eu3+-diagnostic emission lines at 613 - 622 nm to very low signal intensities. In the 325 nm excited spectrum, those Eu3+ emission lines at 613 - 622 nm may be present as weak signals as well as those around 700 nm, but the prominent Dy3+ lines preclude an unequivocal identification. Another Eu3+ sub-level at 685 nm is, similarly, not apparent in the 442 nm excited spectrum next to the intense Sm3+ emission cluster around 700 nm. The weak to missing Eu3+ emission lines in our application example despite suitable excitation conditions with all three employed laser wavelengths in the reference material (see result chapters and summary in Fig. 10) exemplify the difficulties of correct REE identification in natural samples with complex interactions between co-existing REE. The three spectra excited at 325 nm, 442 nm and 532 nm depict effects from energy transfer

between REE in line with low relative emission intensities compared to dominant emission features particularly of Dy3+, Sm3+, Er3+ and Nd3+, and also of Yb3+, Ho3+, Tb3+. How much the lower concentration of Eu3+ compared to the afore-mentioned REE contributes to the low signal intensity is not within the scope of this study."

Review comment:

6. Fig. 11 - I believe that the bands marked in yellow and with a question mark are surely the bands from Sm3 +; in my opinion, the question mark can be removed.

Reply:

Thank you for this advise!

We deleted the question mark at 601 nm and identified this one as Sm emission. We prefer to leave the second question mark at the 872 nm emission as this emission was not evident in our reference data and is not commonly reported for Sm in the literature.

Nevertheless, to the initial discussion of this emission "A change in relative emission intensities is further evident in the 532 nm excited xenotime spectrum at 872 nm (see orange question mark in Fig. 10). One potential candidate responsible for this significant peak could be Nd3+ that has a broad emission cluster in the same range but with opposite intensity ratios."

We added: "Another candidate could be the rarely described near-IR emission of Sm3+ at 872 nm (cf. Wang et al. 2017) that then may be ascribed to a sensitisation of this emission line as a consequence of complex energy transfers between the co-existing REE."